# Kif11-haploinsufficient oocytes reveal spatially differential requirements for chromosome biorientation

Tappei Mishina [1,2], Aurélien Courtois [1,4], Shuhei Yoshida [1], Kohei Asai [1], Hiroshi Kiyonari [3] & Tomoya S Kitajima [1]✉

## Abstract

Bipolar spindle assembly and chromosome biorientation are pre-requisites for chromosome segregation during cell division. The kinesin motor KIF11 (also widely known as Eg5) drives spindle bipolarization by sliding antiparallel microtubules bidirectionally, elongating a spherical spindle into a bipolar-shaped structure in acentrosomal oocytes. During meiosis I, this process stretches homologous chromosome pairs, establishing chromosome biorientation at the spindle equator. The quantitative requirement for KIF11 in acentrosomal spindle bipolarization and homologous chromosome biorientation remains unclear. Here, using a genetic strategy to modulate KIF11 expression levels, we show that *Kif11* haploinsufficiency impairs spindle elongation, leading to the formation of a partially bipolarized spindle during meiosis I in mouse oocytes. While the partially bipolarized spindle allows chromosome stretching in the inner region of its equator, it fails to do so in the outer region, where merotelic kinetochore-microtubule attachments are favored to form. These findings demonstrate the necessity of biallelic functional *Kif11* for bipolar spindle assembly in acentrosomal oocytes and reveal a spatially differential requirement for homologous chromosome biorientation within the spindle.

Keywords Spindle; Oocyte; Meiosis; Chromosome Segregation
Subject Categories Cell Adhesion, Polarity & Cytoskeleton; Cell Cycle

## Introduction

Accurate chromosome segregation during female meiosis is indispensable for the faithful transmission of genetic information to the next generation. Errors in this process result in the production of aneuploid eggs, fertilization of which causes pre-implantation loss, miscarriage, or congenital disorders such as Down syndrome (Herbert et al, 2015; Charalambous et al, 2023).

Chromosome segregation is driven by the spindle, a microtubule-based dynamic machine. In mammalian oocytes, which lack canonical centrosomes, entry to the M-phase of meiosis I promotes microtubule polymerization depending on chromosome-derived diffusible signals (Dumont et al, 2007; Schuh and Ellenberg, 2007; Holubcová et al, 2015; Drutovic et al, 2020). During prometaphase, these microtubules initially assemble into an apolar spherical spindle (Schuh and Ellenberg, 2007). Subsequently, KIF11/Eg5, a plus-end-directed kinesin, crosslinks antiparallel microtubules and slides them bidirectionally (Kapitein et al, 2005). This activity drives bipolar microtubule sorting within the spherical spindle, allowing chromosomes to congress along the spindle surface toward the rim of the future spindle equator, establishing a spatial arrangement of chromosomes called the prometaphase belt (Kitajima et al, 2011). KIF11-mediated microtubule sliding then drives progressive elongation of the spindle, transforming it into a bipolar-shaped structure during the prometaphase-to-metaphase transition (Schuh and Ellenberg, 2007). Spindle elongation allows microtubules to stretch homologous chromosome pairs toward the opposite poles (Schuh and Ellenberg, 2007; Kitajima et al, 2011). Stretched chromosomes align at the spindle equator, establishing chromosome biorientation and forming a spatial arrangement called the metaphase plate. Although both spindle bipolarization and chromosome biorientation are KIF11-dependent processes required for proper chromosome segregation, the spatiotemporal coordination between these processes remains poorly understood.

Chromosome dynamics within the spindle are spatially inhomogeneous. In both centrosomal somatic cells and acentrosomal oocytes, chromosome oscillations are more pronounced in the inner region of the spindle equator (Civelekoglu-Scholey et al, 2013; Takenouchi et al, 2024). In addition, in acentrosomal oocytes, chromosomes moving to the inner region of the spindle equator stretch earlier than those remaining in the outer region during the late prometaphase-to-metaphase transition (Takenouchi et al, 2024). Furthermore, stretched chromosomes in the inner region exhibit increased centromere-to-kinetochore distances during metaphase (Takenouchi et al, 2024), indicative of stronger microtubule-mediated pulling forces. These observations may

[1]Laboratory for Chromosome Segregation, RIKEN Center for Biosystems Dynamics Research (BDR), Kobe, Japan. [2]Faculty of Agriculture, Kyushu University, Fukuoka, Japan. [3]Laboratory for Animal Resources and Genetic Engineering, RIKEN Center for Biosystems Dynamics Research (BDR), Kobe, Japan. [4]Present address: Sex Chromosome Biology Laboratory, The Francis Crick Institute, London, UK. ✉E-mail: tomoya.kitajima@riken.jp

imply that the spindle produces stronger KIF11-dependent bipolar pulling forces in the inner region of its equator. However, it remains unknown whether the amount of KIF11 required for chromosome stretching differs between the inner and outer regions of the spindle equator.

Elucidating the quantitative requirements for KIF11 is critical for advancing molecular understanding underlying female infertility. A recent study has identified heterozygous mutations in the *Kif11* gene in patients experiencing repeated failures of in vitro fertilization or intracytoplasmic sperm injection at fertility clinics (Wu et al, 2024). Overexpression of these mutant forms of KIF11 has been shown to exert dominant-negative effects, disrupting efficient chromosome alignment during meiosis I in human oocytes (Wu et al, 2024). However, whether a heterozygous loss-of-function mutation of *Kif11* impairs acentrosomal spindle bipolarization or homologous chromosome biorientation in oocytes remains unknown.

In this study, we establish a gene knockout strategy to investigate the quantitative requirements for KIF11 in mouse oocytes. We show that heterozygous knockout of the *Kif11* gene in oocytes compromises spindle elongation during the prometaphase-to-metaphase transition of meiosis I, followed by a significant delay or block in anaphase onset. *Kif11*-haploinsufficient oocytes exhibit a partially bipolarized spindle, with chromosomes stretching in the inner region of the metaphase plate but not in the outer region. These results demonstrate that the biallelic expression of functional KIF11 is required for acentrosomal oocyte meiosis and reveal a spatially differential nature of homologous chromosome biorientation on the spindle equator.

# Results and discussion

## Gene knockout strategy to study the quantitative requirements for KIF11 in mouse oocytes

To investigate the quantitative requirements for KIF11 in mouse oocytes, we used a strategy with heterozygous constitutive deletion and meiosis-specific conditional deletion of the *Kif11* gene (Fig. 1A,B). To generate the conditional deletion, we crossed our newly established floxed *Kif11* allele (*Kif11^fl^*, Fig. EV1A) mouse line with the meiosis-specific *Spo11*-Cre recombinase mouse line (Lyndaker et al, 2013). Offspring derived from mice carrying the conditional deletion allele were used to establish a constitutive deletion allele of *Kif11* (*Kif11^del^*). We estimated the expression levels of full-length Kif11 mRNA in isolated fully-grown oocytes using poly(A) RNA sequencing followed by counting of reads corresponding to the first exon of the *Kif11* gene (Fig. 1C). This analysis showed that heterozygous conditional deletion (*Spo11*-Cre, *Kif11^wt/fl^*) reduced Kif11 levels to 59.5%, whereas heterozygous constitutive deletion (*Kif11^fl/del^* without *Spo11*-Cre) reduced them to 22.1% (Fig. 1D). Heterozygous combination of the conditional and constitutive deletion alleles (*Spo11*-Cre, *Kif11^fl/del^*) resulted in a reduction of Kif11 levels to only 3.0% of the wild-type expression levels (Fig. 1D). Comparable levels of depletion were achieved with *Gdf9*-Cre or *Zp3*-Cre (Fig. 1D), two other widely used oocyte-specific drivers (Lewandoski et al, 1997; Vries et al, 2000; Lan et al, 2004). However, further analysis showed that the use of *Gdf9*-Cre or *Zp3*-Cre resulted in residual detectable reads corresponding to

the last exon of the *Kif11* gene, whereas *Spo11*-Cre greatly reduced them (Fig. EV1B). We therefore decided to use *Spo11*-Cre for conditional gene knockout in this study.

To verify the depletion of KIF11 protein, we used a single-oocyte immunofluorescence assay, given the unavailability of antibodies allowing a reliable detection of KIF11 by Western blotting on the limited amount of oocyte extracts. Quantification of immunofluorescence signals at metaphase of meiosis I showed that KIF11 levels on the spindle tended to be reduced in heterozygous conditional deletion oocytes (*Spo11*-Cre, *Kif11^wt/fl^*) and in heterozygous constitutive deletion oocytes (*Kif11^fl/del^* without *Spo11*-Cre) (Fig. EV2A,B), whereas the heterozygous combination of the conditional and constitutive deletion alleles (*Spo11*-Cre, *Kif11^fl/del^*) resulted in a nearly complete depletion of KIF11 signals (Fig. EV2A,B).

We then assessed chromosome alignment on these images. Quantification of the spatial distribution of chromosomes (Fig. EV2C) showed that whereas chromosomes were aligned on the metaphase plate in control oocytes, misaligned chromosomes were significantly increased in heterozygous constitutive deletion (*Kif11^fl/del^* without *Spo11*-Cre) oocytes (Fig. EV2D). Chromosome alignment was never observed in oocytes of the heterozygous combination of conditional and constitutive deletion (*Spo11*-Cre, *Kif11^fl/del^*) (Fig. EV2D). These results suggest that the genetically modified oocytes established in this study serve as models to investigate dose-dependent requirements for KIF11 in acentrosomal oocyte meiosis.

## Acentrosomal spindle elongation requires a full dose of KIF11

To monitor spindle bipolarization, we performed live imaging of the microtubule marker EGFP-MAP4 and the chromosome marker H2B-mCherry throughout M-phase of meiosis I (Schuh and Ellenberg, 2007; Kitajima et al, 2011) (Fig. 2A; Movie EV1). Quantification of spindle shape in 3D showed that control oocytes initially formed an apolar spherical spindle, which gradually elongated into a bipolar-shaped structure (Fig. 2A; Movie EV1), as indicated by a progressive increase in its aspect ratio (Fig. 2B,C). In contrast, spindle elongation was severely perturbed in heterozygous conditional deletion oocytes (*Spo11*-Cre, *Kif11^wt/fl^*) and in heterozygous constitutive deletion oocytes (*Kif11^fl/del^* without *Spo11*-Cre) (Fig. 2A,C; Movie EV1). These oocytes subsequently exhibited a severe delay or block in anaphase entry (Fig. 2D). Oocytes with the heterozygous combination of conditional and constitutive deletions (*Spo11*-Cre, *Kif11^fl/del^*) exhibited almost no detectable spindle elongation or anaphase entry (Fig. 2A,C,D; Movie EV1). Importantly, the spindle elongation defect was rescued by expressing full-length KIF11, but not the mutant form of KIF11 lacking its motor domain, by RNA microinjection into fully-grown oocytes (Fig. EV3A–C). These results demonstrate that efficient spindle elongation and anaphase entry require the biallelic expression of functional KIF11 in acentrosomal oocytes.

We also noticed that KIF11-insufficient oocytes had a significantly smaller spindle volume, which was evident already before the time when control oocytes initiated spindle elongation (2 h after nuclear envelope breakdown, NEBD) (Fig. 2A,E,F; Movie EV1). In contrast, treatment of oocytes with monastrol, an inhibitor of KIF11 motor activity (Kapoor et al, 2000), did not

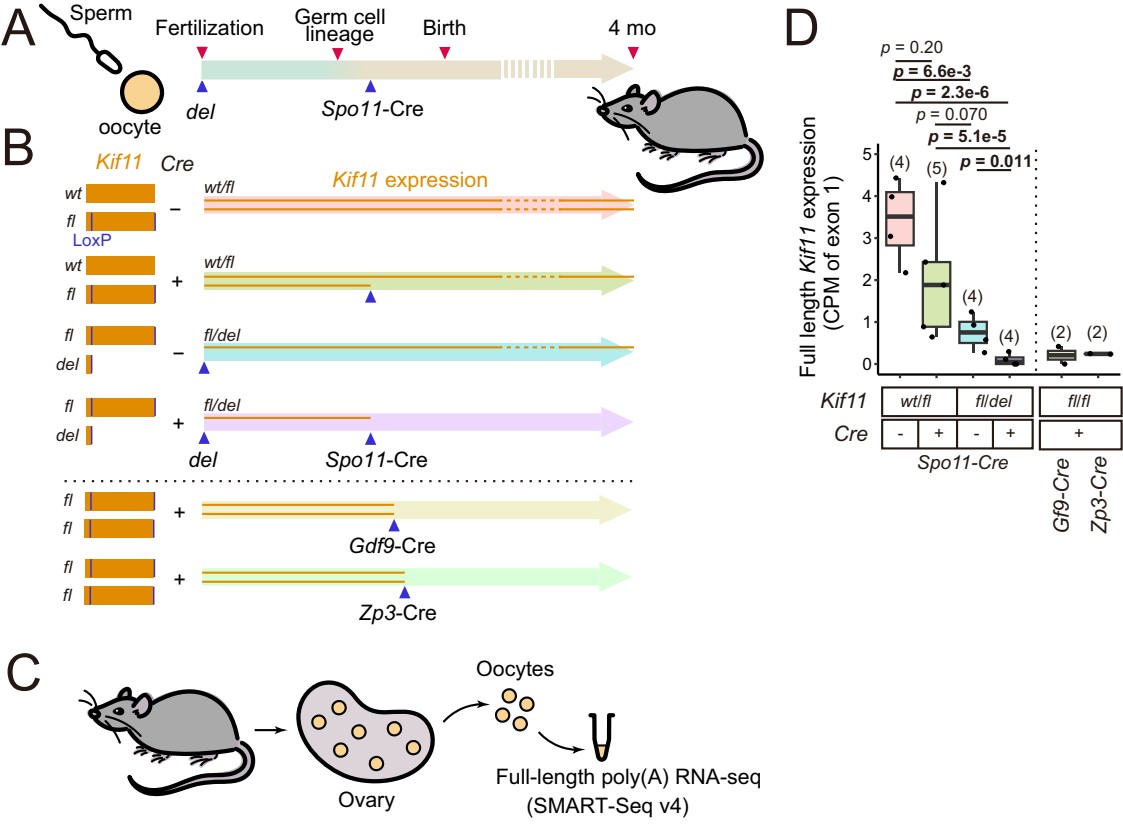

**Figure 1. Gene knockout strategy to study the quantitative requirements for KIF11 in mouse oocytes.**

(A, B) Schematic of the generation of *Kif11* floxed mice. (A) *Spo11-Cre* allows activation of Cre in early prophase. (B) Duration of *Kif11* expression under different genotypes of the combination of *Kif11* (wild-type, floxed, and deleted allele) and *Spo11-Cre*. The orange lines within the arrows indicate the expression of *Kif11*. (C) Design overview of the low-input full-length poly (A) RNA sequencing using SMART-Seq v4 technology. (D) Box plot comparing the relative expression levels of full-length *Kif11* transcripts as measured by the coverage of the first exon from mice oocytes with respect to their genotype (Welch's *t*-test with Holm's correction for multiple comparisons; The numbers in parentheses above the box plot indicate the number of biologically independent experiments). Boxplots: Centerlines indicate the median; box limits represent the 25th and 75th percentiles; whiskers extend to the minimum and maximum values no further than 1.5 * IQR from the hinge (where IQR is the inter-quartile range, or distance between the first and third quartiles). Source data are available online for this figure.

significantly reduce the volume of the microtubule mass (Fig. EV3D–G). These results are consistent with the idea that the KIF11 protein, but not its motor activity, promotes microtubule polymerization in the early stage of acentrosomal spindle formation.

### *Kif11*-haploinsufficient oocytes form a partially bipolarized spindle

Although *Kif11*-haploinsufficient spindles lack an elongated shape, it is still possible that they have bipolar microtubule arrays. To address this, we used mNeonGreen-CEP192, a marker for acentriolar microtubule-organizing centers (MTOCs), which are located at spindle poles where microtubule minus ends are enriched (Clift and Schuh, 2015). As expected, live imaging showed that CEP192-labeled MTOCs were progressively enriched at spindle poles in control oocytes (Fig. 3A; Movie EV2). We found that heterozygous conditional deletion oocytes (*Spo11-Cre, Kif11^{wt/fl}*) also exhibited a bipolar distribution of MTOCs (Fig. 3A–C; Movie EV2), although the distance between the two poles appeared to be smaller than that of control oocytes (Fig. 3D). In heterozygous

constitutive deletion oocytes (*Spo11-Cre, Kif11^{fl/del}*), the bipolar distribution of MTOCs was severely defective, although a weak bipolar enrichment was occasionally observed (Fig. 3A–C; Movie EV2). In oocytes with the heterozygous combination of conditional and constitutive deletions (*Spo11-Cre, Kif11^{fl/del}*), no bipolar MTOC distribution was observed (Fig. 3A–C; Movie EV2). These results suggest that the acentrosomal spindle in *Kif11*-haploinsufficient oocytes, while failing to elongate, manages to establish a partial bipolarity.

### KIF11 insufficiency reveals spatially differential requirement for homologous chromosome biorientation on the spindle equator

These findings led us to investigate how KIF11 insufficiency affects homologous chromosome biorientation. In control oocytes, all homologous chromosome pairs established a stretched state at the spindle equator by metaphase, as indicated by an increased distance between homologous kinetochores (inter-kinetochore distance) (Fig. 4A,B). No spatial bias in inter-kinetochore distance on the spindle equator was observed in control oocytes (Fig. 4C,D),

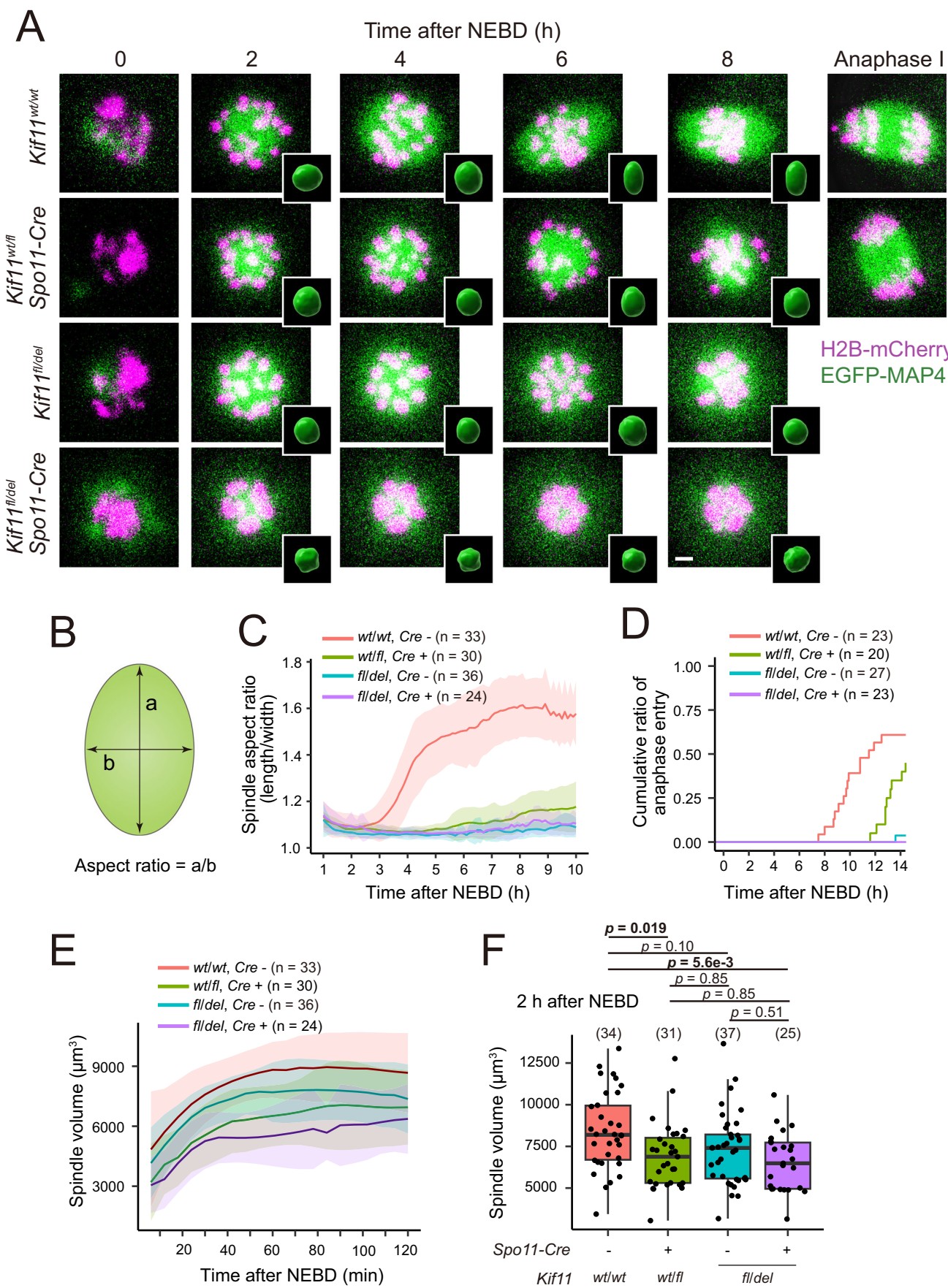

◀  **Figure 2.  Spindle elongation requires a full dose of KIF11.**

(A) Live imaging of *Kif11^{wt/wt}* (control), *Kif11^{wt/fl}* *Spo11-Cre* (single allele depletion from early prophase of meiosis), *Kif11^{fl/del}* (complete single allele depletion), and *Kif11^{fl/del}* *Spo11-Cre* oocytes expressing EGFP-MAP4 (microtubules, green) and H2B-mCherry (chromosomes, magenta). Z-projection images are shown. 3D-reconstructed spindles are shown in insets. Scale bars 5 μm for z-projection images but not for 3D-reconstructed images. (B) Diagram for measuring the aspect ratio (length/width) of the 3D-reconstructed spindle. (C) Quantification of spindle bipolarization is shown as mean ± SD. Aspect ratio was measured. (D) Timing of anaphase entry for each genotype. (E) Spindle volume over time with mean ± SD. Spindle volume was determined from 3D-reconstructed images. (F) Box plot comparing spindle volume at prometaphase (2 h after nuclear envelope breakdown, NEBD) between genotypes (Welch's *t*-test comparing all possible pairs with Holm's correction for multiple comparisons). Three (C, E, F) or two (D) biologically independent experiments were performed. Boxplots: Centerlines indicate the median; box limits represent the 25th and 75th percentiles; whiskers extend to the minimum and maximum values no further than 1.5 * IQR from the hinge (where IQR is the inter-quartile range, or distance between the first and third quartiles). Source data are available online for this figure.

consistent with our previous report (Takenouchi et al, 2024). Interestingly, we found that in heterozygous conditional deletion oocytes (*Spo11-Cre, Kif11^{wt/fl}*), homologous chromosome pairs in the inner region of the spindle equator were stretched with an increased inter-kinetochore distance, indicative of chromosome biorientation, while homologous chromosome pairs in the outer region were not (Fig. 4A–D). In heterozygous constitutive deletion oocytes (*Spo11-Cre, Kif11^{fl/del}*), most homologous chromosome pairs were not stretched and were preferentially located in the outer region of the spindle equator (Fig. 4A–D). These results suggest that a higher dose of KIF11 is required for homologous chromosome biorientation at the outer region of the spindle equator in acentrosomal oocytes.

## Merotelic kinetochore-microtubule attachments prefer to form in the outer region of the spindle equator

To gain insight into the spatial difference in the requirement for chromosome stretching, we visualized kinetochore-microtubule attachments (Fig. 4E). In control oocytes, where all chromosomes were aligned and fully stretched in both the outer and inner regions of the spindle equator (Fig. EV4A–D), kinetochores attached to microtubules from opposite poles (merotelic attachment) were preferentially found in the outer region (Fig. 4F). Notably, a similar spatial bias of merotelic attachment to the outer region was observed in heterozygous conditional deletion oocytes (*Spo11-Cre, Kif11^{wt/fl}*) (Fig. 4F), where most chromosomes were not stretched in the outer region (Fig. EV4A–D). These results suggest that the spindle prefers to form merotelic kinetochore-microtubule attachments in the outer region of the equator, regardless of the stretched or unstretched state of chromosomes. We propose that the full dose of KIF11 ensures chromosome stretching with merotelic kinetochores in the outer region of the spindle equator.

## Concluding remarks

In this study, we established mouse oocyte models with insufficient expression levels of KIF11, demonstrating that *Kif11* haploinsufficiency causes severe defects in acentrosomal spindle elongation during female meiosis I. The delay in spindle elongation in *Kif11*-haploinsufficient oocytes resulted in chromosome biorientation defects, particularly at the outer region of the spindle equator. These results suggest a spatially differential requirement for homologous chromosome biorientation on the spindle equator in oocytes: while the reduced levels of KIF11 are sufficient for homologous chromosome biorientation in the inner region, a full dose of KIF11 is required in the outer region (Fig. 5).

We propose two possible explanations for the spatially differential requirement for homologous chromosome biorientation. One possibility is that the polar ejection force is stronger in the outer region of the spindle equator. This force, which pushes chromosomes from the spindle poles toward the equator, can counteract KIF11-dependent bipolar microtubule pulling forces required for chromosome biorientation (Rieder et al, 1986; Chong et al, 2024). Stronger polar ejection forces in the outer region of the spindle equator may also explain less pronounced chromosome oscillation, as suggested in studies using centrosomal somatic cells (Civelekoglu-Scholey et al, 2013).

Alternatively, factors that cooperate with KIF11 to promote chromosome biorientation may be more concentrated in the inner region of the spindle equator. It is possible that the acentrosomal spindle has denser microtubules in its inner region prior to bipolarization, which may subsequently promote KIF11-mediated chromosome biorientation. Being occupied and surrounded by chromosomes, the inner region may experience higher concentrations of chromosome-derived signals, creating a microenvironment that enhances KIF11-mediated chromosome biorientation. For instance, chromosomes produce diffusible RanGTP signals that activate spindle assembly factors (Carazo-Salas et al, 1999; Kalab et al, 2002) such as HURP, a bundling factor that localizes to kinetochore-attached microtubules (Silljé et al, 2006; Wong and Fang, 2006; Koffa et al, 2006). HURP accumulates more prominently in the inner region of the spindle equator (Takenouchi et al, 2024), potentially cooperating with KIF11 to facilitate chromosome biorientation. Given that chromosomal RanGTP signals are more essential for spindle assembly in acentrosomal oocytes than in centrosomal somatic cells (Dumont et al, 2007; Schuh and Ellenberg, 2007; Moutinho-Pereira et al, 2013; Holubcová et al, 2015; Drutovic et al, 2020), the spatially differential nature of chromosome biorientation may be specific to acentrosomal oocytes. It is also possible that the meiosis I-specific chromosome feature contributes to the spatially differential nature of biorientation, as homologous chromosome biorientation during meiosis I requires kinetochore pair separation over greater distances compared to sister chromatid biorientation during meiosis II and mitosis.

In either scenario, the spindle likely exerts stronger bipolar pulling forces on chromosomes in the inner region of its equator. On the other hand, pulling forces are relatively weaker in the outer region, which is associated with increased opportunities for merotelic kinetochore-microtubule attachment. We suggest that this spatial difference makes chromosome stretching in the outer region particularly challenging and therefore sensitive to a reduction in KIF11 dosage.

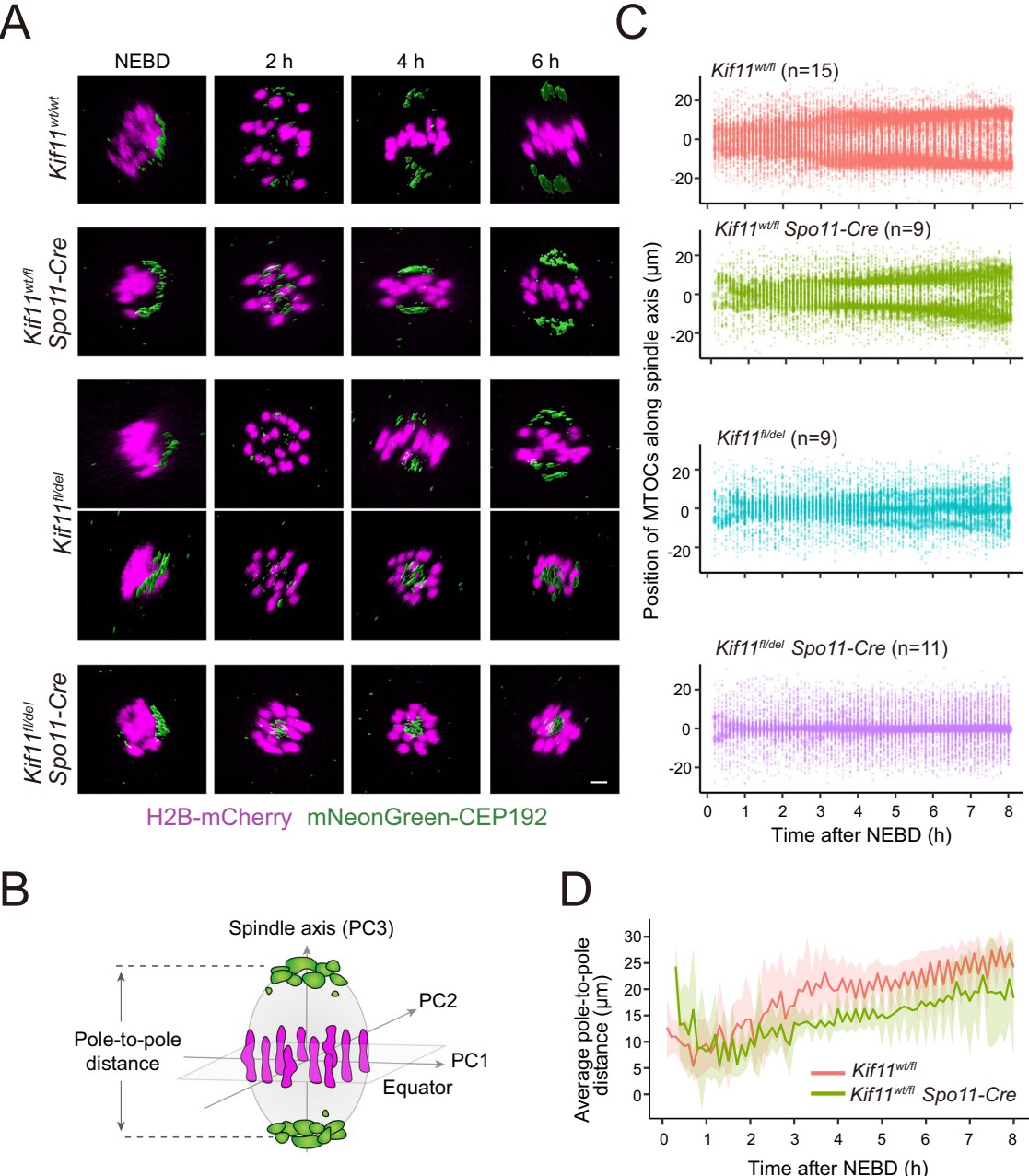

**Figure 3. *Kif11*-haploinsufficient oocytes form a partially bipolarized spindle.**

(A) Live imaging oocytes expressing mNeonGreen-CEP192 (MTOCs, green; surface rendered) and H2B-mCherry (chromosomes, magenta) for each *Kif11* deletion genotype. 3D-reconstructed images with signal interpolation in z are shown. Scale bars 5 μm. (B) Diagram showing determination of the spindle axis and equator and the measurement of pole-to-pole distances based on chromosome and MTOC positions. (C) Bipolar MTOC sorting depends on KIF11 dosage. MTOC volume and its position along the spindle axis over time were measured from 3D-reconstructed images. The dots represent each MTOC with its size scaled by the relative MTOC volume. (D) Line plot comparing the two pole-to-pole distances between *Kif11^wt/fl^* and *Kif11^wt/fl^ Spo11-Cre* oocytes. Two biologically independent experiments were performed. Source data are available online for this figure.

The idea of spatially differential pulling forces aligns with our recent findings that the inner region of the metaphase plate has an increased risk of premature chromosome separation, a process depending on bipolar microtubule pulling forces, in aged oocytes (Takenouchi et al, 2024). Premature chromosome separation is a major cause of aging-associated chromosome segregation errors in oocytes (Sakakibara et al,

2015; Zielinska et al, 2015). Identifying mechanisms that regulate the spatial distribution of bipolar forces in the spindle will contribute to our understanding of the cause of egg aneuploidy.

Additionally, this study demonstrates severe meiotic defects caused by *Kif11* haploinsufficiency in oocytes. Heterozygous mutations in the *Kif11* gene have been identified in infertile

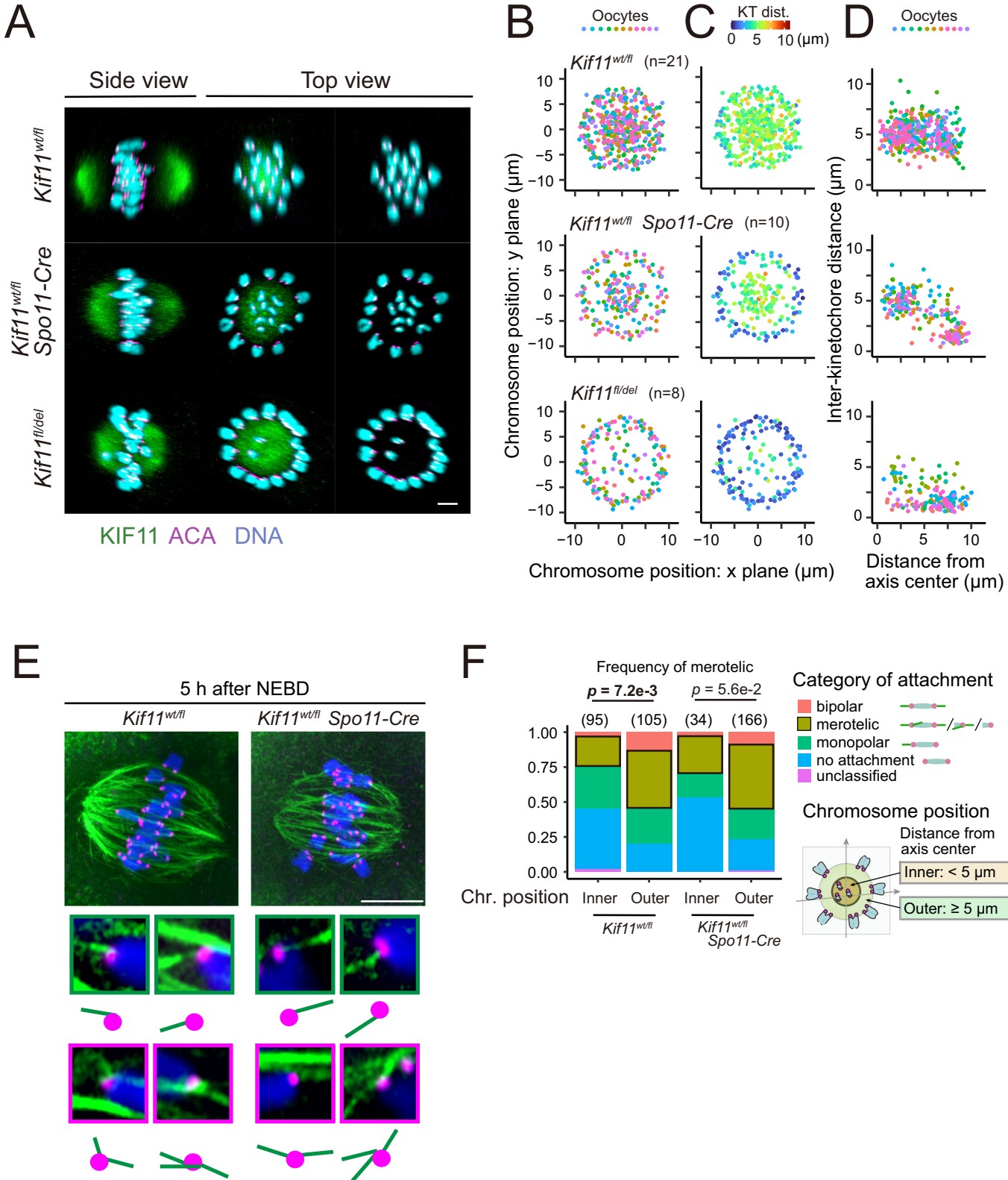

**Figure 4. Spatially differential requirement for chromosome biorientation on the spindle equator.**

(A) Representative images of chromosome position and KIF11 localization for each genotype. Images are viewed in 3D with signal interpolation in z. Scale bar, 5 μm. (B) Distribution of chromosome positions on the spindle equator is shown. Each dot represents a chromosome. Colors indicate oocytes. (C) Chromosome positions are shown as in B, with color codes indicating inter-kinetochore distance. (D) 2D plot of inter-homologous kinetochore distance with respect to its position from the center of the spindle equator (right). Colors indicate oocytes. (E) Representative image of stable end-on kinetochore-microtubule (KT-MT) attachments for *Kif11*$^{wt/fl}$ and *Kif11*$^{wt/fl}$ *Spo11-Cre* oocytes. Cold-treated oocytes were stained for MTs (green), KTs (ACA, magenta), and chromosomes (Hoechst 33342, blue). Magnified images of representative end-on merotelic (magenta frame) and correct end-on (green frame) attachments are shown. Contrasts were adjusted in each of the magnified images. Scale bar, 10 μm. (F) Frequency of KT-MT attachment classification showing frequent merotelic KT-MT attachments in chromosomes located in the outer region of the spindle. Chromosome KT-MT attachment status was classified as "bipolar" (correct end-on at both KTs), "merotelic" (one or both of the KTs with end-on from both spindle poles), "monopolar" (correct end-on at one of the KTs), "no attachment" at both KTs, and "unclassified". Chromosomes were defined as "inner" or "outer" chromosomes if they were located <5 or ≥5 μm, respectively, from the center of the spindle equator. The numbers in parentheses above the bar indicate the number of kinetochores included in each category. The statistically significant differences in the frequency of merotelic attachment between "inner" and "outer" regions were compared for each genotype using Fisher's exact with Holm's correction for multiple comparisons. Two and three biologically independent experiments were performed for panels (A–F), respectively. Source data are available online for this figure.

patients at fertility clinics (Wu et al, 2024). Although several mutant forms of KIF11 have dominant-negative effects when overexpressed (Wu et al, 2024), our data suggest that a heterozygous loss-of-function of *Kif11* results in severe meiotic defects in oocytes and thus female infertility. Understanding the dosage-dependent role of KIF11 is therefore critical for advancing the molecular understanding of infertility and improving assisted reproductive technologies.

# Methods

### Reagents and tools table

| Reagent/resource | Reference or source | Identifier or catalog number |
|---|---|---|
| **Experimental models** | | |
| C57BL/6NCrSlc (*M. musculus*) | Japan SLC | |
| B6D2F1/Slc (*M. musculus*) | Japan SLC | (C57BL/6NCrSlc ♀ × DBA/2CrSlc ♂) F1 |
| Tg(Spo11-cre)1Rsw (*M. musculus*) | The Jackson Laboratory (Lyndaker et al, 2013) | 032646 |
| Tg(Gdf9-icre)5092Coo (*M. musculus*) | The Jackson Laboratory (Lan et al, 2004) | 011062 |
| Tg(Zp3-cre)3Knw(*M. musculus*) | The Jackson Laboratory (de Vries et al, 2000) | 003651 |
| *Kif11* flox (*M. musculus*) | This study | |
| *Kif11*$^{flox/flox}$ Zp3-Cre (*M. musculus*) | This study | |
| *Kif11*$^{flox/flox}$ Gdf9-Cre (*M. musculus*) | This study | |
| **Recombinant DNA** | | |
| pGEMHE_EGFP-Map4 | Schuh and Ellenberg 2007 | |
| pGEMHE_H2B-mCherry | Kitajima et al, 2011 | |
| pGEMHE_mNeonGreen-Cep192 | Kyogoku and Kitajima 2017 (originally Clift and Schuh 2015) | |
| pGEMHE_mCherry | Asai et al, 2024 | |
| pGEMHE_Kif11-mCherry | This study | |

| Reagent/resource | Reference or source | Identifier or catalog number |
|---|---|---|
| pGEMHE_Kif11Δmotor-mCherry | This study | |
| pGEMHE_H2B-SNAP | Takenouchi et al, 2024 | |
| **Antibodies** | | |
| human anti-centromere protein antibody | Antibodies Incorporated | 15-234 |
| rabbit anti-Kif11 | Sigma | HPA010568 |
| rat monoclonal anti-alpha tubulin | Bio-Rad | MCA77G |
| Alexa Fluor 488 goat anti-rabbit IgG (H + L) | Thermo Fisher | A11034 |
| Alexa Fluor 488 goat anti-rat IgG (H + L) | Thermo Fisher | A11006 |
| Alexa Fluor 555 goat anti-human IgG (H + L) | Thermo Fisher | A21433 |
| **Oligonucleotides and other sequence-based reagents** | | |
| custom ssODN | This study; Fasmac | Figure EV1 |
| crRNA | Fasmac | GE-001 |
| tracrRNA | Fasmac | GE-002 |
| **Chemicals, enzymes and other reagents** | | |
| Equine chorionic gonadotropin | ASKA Pharmaceutical | G4877-2000IU |
| CARD HyperOva | KYUDO | F-021 |
| mMESSAGE mMACHINE T7 kit | invitrogen | AM1344 |
| 3-isobutyl-1-methyl-xanthine | Sigma-Aldrich | 15879 |
| monastrol | Sigma-Aldrich | M815 |
| EcoRV | New England Biolabs | R0195S |
| SNAP-Cell 647-SiR | New England Biolabs | S9102S |
| Hoechst 33342 | Invitrogen | H3570 |
| SMART-Seq v4 Ultra Low Input RNA Kit for Sequencing | Clontech | 634891 |
| Nextera XT DNA Library Preparation Kit | Illumina | FC-131-1096 |

| Reagent/resource | Reference or source | Identifier or catalog number |
|---|---|---|
| Nextera XT Index Kit | Illumina | FC-131-1001 |
| SPRISelect kit | Beckman Coulter | B23318 |
| **Software** | | |
| Fiji | https://fiji.sc/ | |
| Imaris | Oxford Instruments | |
| base R (v4.3.0) | R Foundation for Statistical Computing | |
| fastp | Chen et al, 2018 | |
| HiSat2 v2.2.1 | Kim et al, 2019 | |
| SAMtools | Danecek et al, 2021 | |
| featureCounts v2.0.3 | Liao et al, 2014 | |
| DESeq2 | Love et al, 2014 | |
| **Other** | | |
| Illumina HiseqX | Illumina | |
| LSM780, 880 confocal microscope | Zeiss | |

## Methods and protocols

### Mouse

The C57BL/6 background mice were used for genetic engineering. *Spo11-Cre* (Lyndaker et al, 2013), *Gdf9-Cre* (Lan et al, 2004), and *Zp3-Cre* (Vries et al, 2000) mice were obtained from the Jackson Laboratory (Strain #032646, #011062, and #003651, respectively). BDF1 mice were used for the experiments shown in Fig. EV3D–G. All mouse experiments were approved by the Institutional Animal Care and Use Committee at RIKEN Kobe Branch (IACUC).

### Generation of Kif11 floxed mice

The *Kif11* conditional knockout (floxed) mice (Accession No.: CDB0012E: https://large.riken.jp/distribution/mutant-list.html) were generated by CRISPR/Cas9-mediated genome editing in C57BL/6 zygotes using single-strand oligodeoxynucleotides (ssODN) as previously described (Hashimoto et al, 2016). The gRNA sites were designed using CRISPRdirect (Naito et al, 2014), and crRNA/tracrRNA and ssODN were chemically synthesized (Fasmac Co., Ltd). The floxed alleles are depicted in Fig. EV1A. Genotyping PCR was performed using the following primers, followed by EcoRV digestion: 5gtFW (5′-CTCCCGGTTCTCACTGTGTC-3′) and 5gtREV (5′-TGCACCT-TAGCCATGTACTTTCA-3′) (WT: 731 bp, 5′-loxP: 323 and 446 bp) and 3gtFW(5′-GGCCAAGGCTGTTTCCCTAC-3′) and 3gtREV(5′-ACAGCGTTGTCAAAGCGAAA-3′) (WT: 500 bp, 3′-loxP: 322 and 218 bp). *Kif11* floxed mice were mated with *Spo11-Cre*, *Gdf9-Cre*, or *Zp3-Cre* mice to generate conditional knock-out.

### Mouse oocyte culture

Female mice of three to four months old were injected with 5 IU of equine chorionic gonadotropin (eCG, ASKA Pharmaceutical) or 0.1 mL of CARD HyperOva (KYUDO) to hyperovulation. Full-grown oocytes at the germinal vesicle (GV) stage were collected from the ovaries 48 h after injection. The isolated oocytes were cultured in M2 medium containing 200 μM 3-isobutyl-1-methyl-xanthine (IBMX, Sigma) at 37 °C. Meiotic resumption was induced by washing to remove IBMX. When indicated, 100 μM monastrol was used.

### Transcriptome analysis

Low-input RNA sequencing was conducted using the SMART-seq v4-based method following the manufacturer's protocol with slight modifications in a 1/4 volume reaction. Briefly, isolated full-grown oocytes at the germinal vesicle (GV) stage were washed in washing

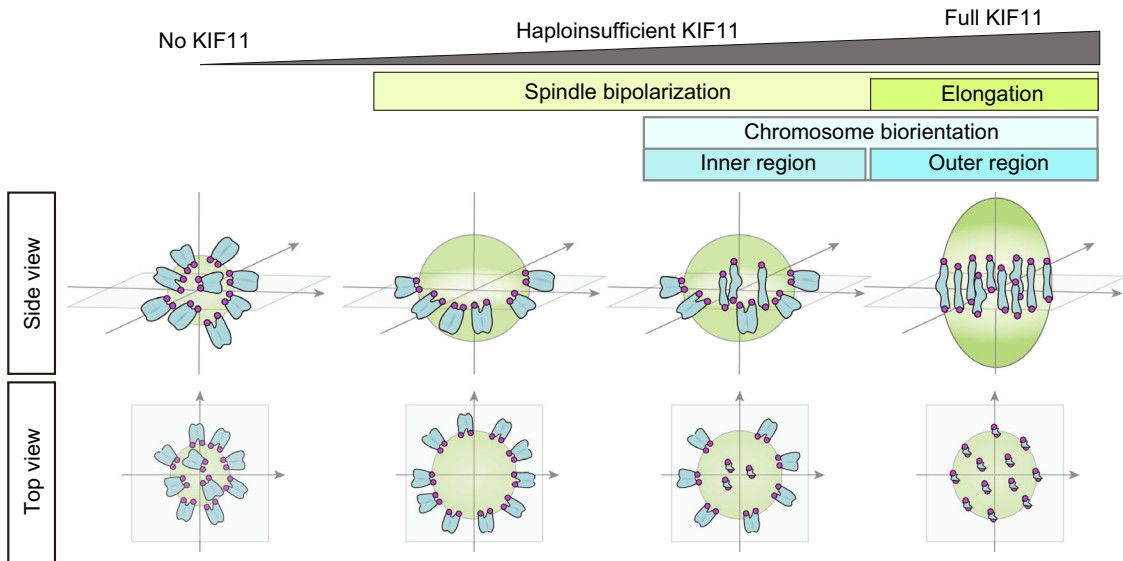

**Figure 5. Model for spatially differential requirement of KIF11 dose for chromosome biorientation within the spindle.**

Diagram of spatially differential requirements of KIF11 during oocyte meiosis I. In oocytes with haploinsufficient KIF11, spindle elongation is impaired, leading to the formation of a partially bipolarized spindle, where chromosomes stretch in the inner region of the spindle equator but fail to do so in the outer region.

medium [0.1% PBA in PBS] and collected manually with 0.5 μL of the washing medium in 2.125 μl cell lysis buffer [10X Lysis Buffer (TAKARA), RNase Inhibitor (TAKARA), and RNase-free water (Nacalai)] in 0.2 mL 8-strip tubes. Three oocytes were collected in each tube with one or two replicates from each mouse. The collected oocytes were lysed and stored at −80 °C until library preparation.

The SMART-seq v4 Ultra Low Input RNA Kit for Sequencing (Takara) was used to reverse transcribe poly(A) RNA and amplify full-length cDNA. Samples were amplified for 13 cycles in eight-strip tubes. After the cDNA was adjusted to 0.5 ng in 1.25 μL of elution buffer [10 mM Tris-HCl, pH 8.5], the Tn5 tagmentation-based reaction was performed with 1/4 volume of the Nextera XT DNA Library Preparation Kit (Illumina) with Nextera XT Index Kit (Illumina) according to the manufacturer's protocol. Library DNA was amplified with 12 cycles of PCR and purified using 1.8× volume of SPRISelect (Beckman Coulter) and eluted into 12 μL of the elution buffer [10 mM Tris-HCl, pH 8.5]. Libraries were sequenced using HiSeqX with 150-bp paired-end. In total, 17 libraries were sequenced.

Hisat2 v2.2.1 (Kim et al, 2019) was used to align the reads to the mouse genome (GRCm38) after trimming adapter sequences and low-quality bases using fastp (Chen et al, 2018) with the option "-3 -q 15 -l 15". The resulting binary alignment/map (BAM) files were sorted using SAMtools (Danecek et al, 2021). The featureCounts v2.0.3 tool implemented in the Subread software (Liao et al, 2014) was used to generate counts of reads uniquely mapped to annotated genes using the Ensembl (release 93) annotation gtf file. In order to quantify the first exon of Kif11, as a proxy for the full-length mRNA, reads were counted using the custom annotation gtf file with all Kif11 exons removed except for the exon 1. Differential expression of Kif11 between genotypes was tested with the DESeq2 package (Love et al, 2014) in R, which normalizes library sizes using the relative log expression (RLE) method, using the dataset of genes expressed in at least 3 samples. All possible genotype pairs under the *Spo11-Cre* +/− background were compared with Holm's correction for multiple comparisons.

### Live imaging

Messenger RNAs were transcribed in vitro using the mMESSAGE mMACHINE T7 Kit (Invitrogen). The following mRNAs were introduced by microinjection into fully-grown GV-stage mouse oocytes: 0.4–0.6 pg H2B-mCherry (Kitajima et al, 2011), 2.4–7 pg EGFP-Map4 (Schuh and Ellenberg, 2007), 0.6 pg H2B-SNAP (Takenouchi et al, 2024), 1.5 pg Kif11-mCherry, 1.5 pg Kif11$^{\Delta Motor}$-mCherry, 1.5 pg mCherry (Asai et al, 2024), and 0.4–0.6 pg mNeonGreen-Cep192 (Clift and Schuh, 2015; Kyogoku and Kitajima, 2017). For SNAP imaging, the oocytes were cultured in medium containing SNAP-Cell® 647-SiR (New England Biolabs) for 1 h at 37 °C. The oocytes were released from the dyes before live imaging. Live imaging was performed with a Zeiss LSM880 or LSM780 confocal microscope equipped with a 40x C-Apochromat 1.2NA water immersion objective lens, controlled by MyPiC (Politi et al, 2018). For spindle and chromosome imaging of wild-type and cKO mouse oocytes, 14–16 confocal z-sections (every 3 μm) of 512 × 512 pixel xy images were acquired every 5–6 min, whereas for control (DMSO) and inhibitor (Monastrol) treated oocyte imaging, 25 confocal z-sections (every 1.5 μm) of 256 × 256 pixel xy images were acquired at 5 min intervals. For microtubule-organizing center (MTOC) imaging, 25 confocal z-sections (every 15 μm) of 512 × 512 pixel xy images were acquired every 12 min.

### 4D image analysis

Spindle and MTOCs were reconstructed into 3D surface renderings of EGFP-MAP4 and mNeonGreen-CEP192 signals, respectively, using Imaris software (Oxford Instruments). For each time point, the generated 3D surfaces were used to calculate the volume and its position (xyz coordinate) by the center of mass. For spindle shape analysis, the aspect ratio was determined by the length to width of an ellipsoid fitted to the generated 3D surface. To generate kymographs of MTOCs, chromosome positions were recorded in 3D for each time point. Principal component analysis (PCA) of chromosome positions was then performed to define the new xyz coordinates represented by the plate (PC1 and PC2 coordinates) and spindle axis (PC3 coordinates), as well as the magnitude of explained variance of each principal axis contribution (square of standard deviation; for example, complete chromosome alignment converged to explained variance of PC1, PC2, and PC3 axis as 0.5, 0.5, and 0, respectively). The 3D-reconstructed MTOCs positions were then transformed based on the new coordinates. Images covering almost the entire spindle were used for the analysis, and others were excluded.

### Immunostaining

Oocytes were fixed with 1.6% formaldehyde in 10 mM PIPES (pH 7.0), 1 mM MgCl₂, and 0.1% Triton X-100 for 30 min, followed by permeabilization with PBT (PBS supplemented with 0.1% Triton X-100) at 4 °C overnight. When indicated, oocytes were pretreated with cold M2 medium on ice for 5 min (Figs. 4E,F and EV4) before fixation. The oocytes were blocked with 3% bovine serum albumin (BSA)-PBT for 2 h and incubated at 4 °C overnight with primary antibodies. The oocytes were washed with 3% BSA-PBT and then incubated with secondary antibodies and 5 μg/ml Hoechst 33342 for 2 h or overnight (Figs. 4E,F and EV4). The oocytes were washed again and suspended in 0.01% BSA-PBS. The oocytes were imaged under a Zeiss LSM780 confocal microscope equipped with a GaAsP detector or an LSM880 confocal microscope with AiryScan (Figs. 4E,F and EV4) and a 40x C-Apochromat 1.2NA water immersion objective lens. We recorded z-confocal sections (every 1 μm) of 512 × 512 pixel xy images to capture the entire spindle of the oocyte using LSM780, or z-confocal sections (every 0.2 μm) of 1000 × 1000 pixel xy images to capture the spindle microtubules and kinetochores using LSM880 (Figs. 4E,F and EV4).

The following primary antibodies were used: human anti-centromere antibodies (1:200, ACA, 15-234, Europa Bioproducts), a rabbit anti-KIF11 (1:500, HPA010568, Sigma), and a rat monoclonal anti-alpha tubulin (1:2000, MCA77G, Bio-Rad). The secondary antibodies were Alexa Fluor 488 goat anti-rabbit IgG (H + L) (A11034); Alexa Fluor 555 goat anti-human IgG (H + L) (A21433); Alexa Fluor 488 goat anti-rat IgG (H + L) (A11006); Alexa Fluor 555 goat anti-human IgG (H + L) (A21433). Representative single slice or Z-projection images are shown.

### Quantification of fluorescent signal intensity

Fiji (https://fiji.sc/) was used to quantify fluorescent signals. To determine the KIF11 levels at the spindle pole, the mean fluorescence intensity of KIF11 around the peak of the signal was measured. For *Kif11$^{fl/del}$ Spo11-Cre* oocytes, KIF11 was quantified at the chromosome centers based on the microtubule enrichment in live imaging. The levels of ACA at kinetochores were also

measured. The mean background intensity was obtained in the cytoplasmic region. The ratio of the value obtained from the KIF11 to that of ACA was calculated.

For the chromosome and kinetochore distribution analysis, images were reconstructed in 3D with Imaris, and the 40 kinetochore positions were manually determined. Chromosome positions were represented as the center of its two homologous kinetochores. The degree of chromosome alignment was calculated by performing the PCA on the xyz coordinates of the 20 chromosome positions as in the MTOCs analysis above. Oocytes were considered to have "aligned chromosomes" if their cumulative explained variance of PC1 and PC2 for chromosome positions was >0.90.

For the kinetochore-microtubule (KT-MT) attachment analysis, the chromosome KT-MT attachment status was manually classified as "bipolar" (correct end-on at both KTs), "merotelic" (one or both of the KTs with end-on from both spindle poles), "monopolar" (correct end-on at one of the KTs), "no attachment" at both KTs, and "unclassified" when kinetochores are located near the acentriolar microtubule-organizing center, where dense microtubules interfere with classification. Chromosomes were defined as "inner" or "outer" chromosomes if they were located <5 or ≥5 μm, respectively, from the center of the spindle equator.

### Statistical analysis

Statistical significance was examined with R. Statistical tests, sample sizes and $p$ values are shown in figures and figure legends. When needed, Holm's corrections were applied for correction of multiple comparisons.

## Data availability

Microscopy images acquired and analyzed in this study were deposited in the BioImage Archive under accession number S-BIAD1946. Expression profile data generated and analyzed in this study were deposited in the NCBI Gene Expression Omnibus (https://www.ncbi.nlm.nih.gov/geo/query/acc.cgi?acc=GSE284383) database under accession number GSE284383.

The source data of this paper are collected in the following database record: biostudies:S-SCDT-10_1038-S44319-025-00539-w.

## Peer review information

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

## Acknowledgements

We thank J. Ellenberg for a microscope automation macro, the imaging and animal facilities of RIKEN BDR for technical support, the Center for Advanced Technical and Educational Supports, Faculty of Agriculture, Kyushu University, for part of the image analysis, and all lab members for discussions and comments. TM was supported by the RIKEN Special Postdoctoral Researchers Program. This work was supported by RIKEN intramural grants, RIKEN Pioneering Project "Long-Timescale Molecular Chronobiology", JSPS KAKENHI Grant Number 23H04948/21H02407/25H00981, and Naito Foundation Research Grant to TSK.

## Author contributions

**Tappei Mishina**: Conceptualization; Data curation; Formal analysis; Investigation; Visualization; Methodology; Writing—original draft; Writing—review and editing. **Aurélien Courtois**: Conceptualization; Resources; Investigation; Methodology; Writing—original draft. **Shuhei Yoshida**: Data curation; Formal analysis; Investigation; Visualization; Methodology; Writing—original draft. **Kohei Asai**: Data curation; Formal analysis; Investigation; Visualization; Writing—original draft. **Hiroshi Kiyonari**: Resources; Investigation; Writing—original draft; Project administration. **Tomoya, S Kitajima**: Conceptualization; Supervision; Funding acquisition; Visualization; Writing—original draft; Project administration; Writing—review and editing.

Source data underlying figure panels in this paper may have individual authorship assigned. Where available, figure panel/source data authorship is listed in the following database record: biostudies:S-SCDT-10_1038-S44319-025-00539-w.

## Disclosure and competing interests statement

The authors declare no competing interests.

# Expanded View Figures

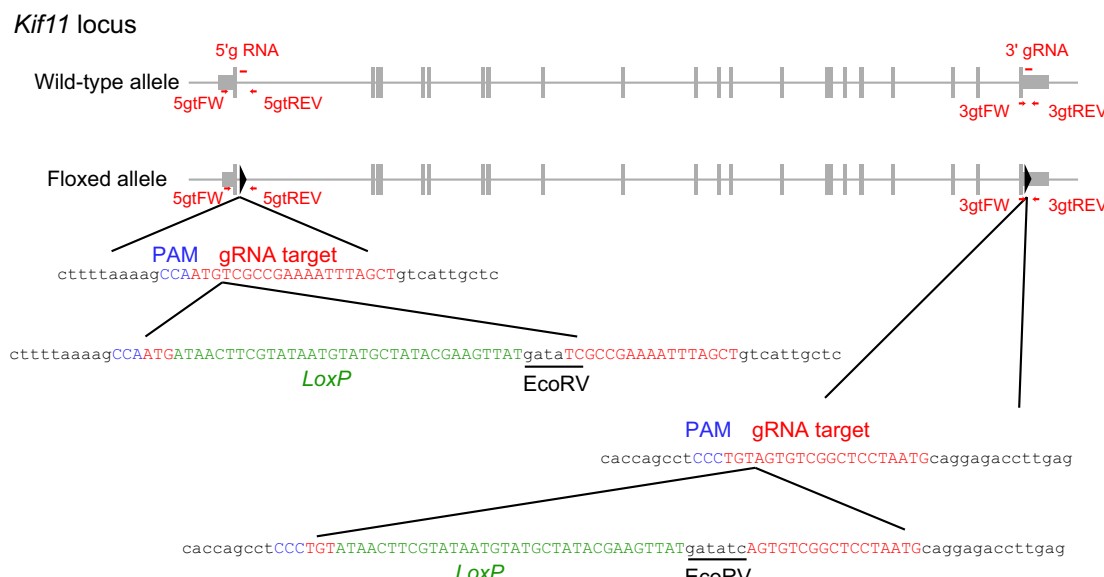

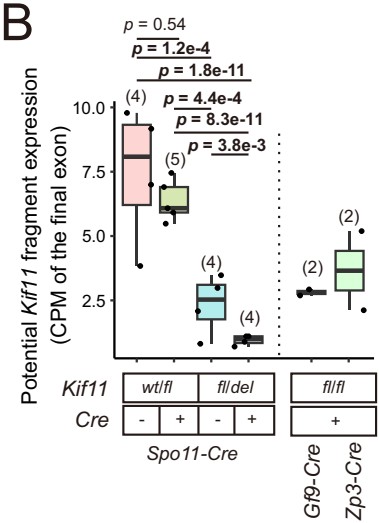

**Figure EV1. Generation of *Kif11* floxed mice.**

(**A**) Diagram of the *Kif11* gene locus. Thin exon segments denote untranslated regions (UTR), and the thick exon segments denote coding regions. Black triangles indicate target sites for CRISPR-Cas9-mediated *Loxp* (green) insertion. Red arrows indicate primer positions used for genotyping. The protospacer adjacent motif (PAM, blue) and the guide RNA (gRNA) target sequences (red) are marked. (**B**) Box plot comparing the relative expression levels of residual *Kif11* transcripts as measured by the coverage of the final exon from mice oocytes with respect to their genotype (Welch's *t*-test with Holm's correction for multiple comparisons; The numbers in parentheses above the box plot indicate the number of biologically independent experiments). Boxplots: Centerlines indicate the median; box limits represent the 25th and 75th percentiles; whiskers extend to the minimum and maximum values no further than 1.5 * IQR from the hinge (where IQR is the inter-quartile range, or distance between the first and third quartiles). Source data are available online for this figure.

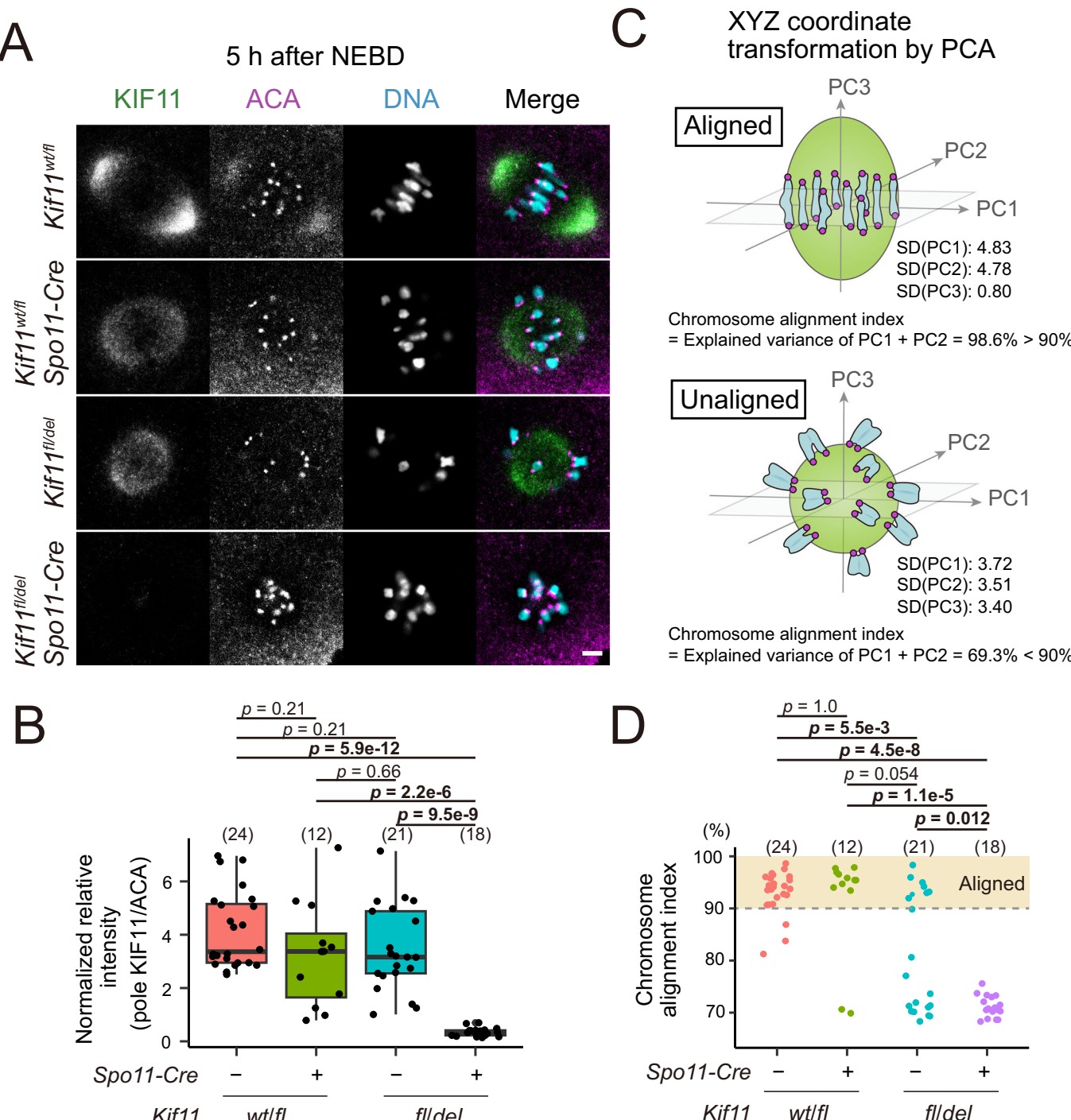

**Figure EV2.  Validation of KIF11 depletion.**

(A) Oocytes were stained for KIF11 (green), ACA (kinetochores, magenta), and Hoechst 33342 (DNA, cyan) fixed at 5 h after NEBD (metaphase). Single z-section images are shown. Scale bar, 5 μm. (B) Quantification of KIF11 accumulated at the spindle pole. KIF11 signals in *Kif11$^{fl/del}$ Spo11-Cre* oocytes were quantified at the chromosome centers. Welch's *t*-test comparing all possible pairs with Holm's correction for multiple comparisons. (C, D) Chromosome alignment defects in *Kif11*-deleted oocytes. Chromosome alignment index, defined as the sum of explained variance of PC1 and PC2 from PCA-transformed coordinates of chromosome position in 3D images, greater than 90% were considered as "chromosome-aligned" oocytes. The statistically significant differences (*p* < 0.05) in the frequency of chromosome-aligned oocytes between genotypes were compared by Fisher's exact test comparing all possible pairs with Holm's correction for multiple comparisons. Two biologically independent experiments were performed for panels (A, B, D). Boxplots: Centerlines indicate the median; box limits represent the 25th and 75th percentiles; whiskers extend to the minimum and maximum values no further than 1.5 * IQR from the hinge (where IQR is the inter-quartile range, or distance between the first and third quartiles). Source data are available online for this figure.

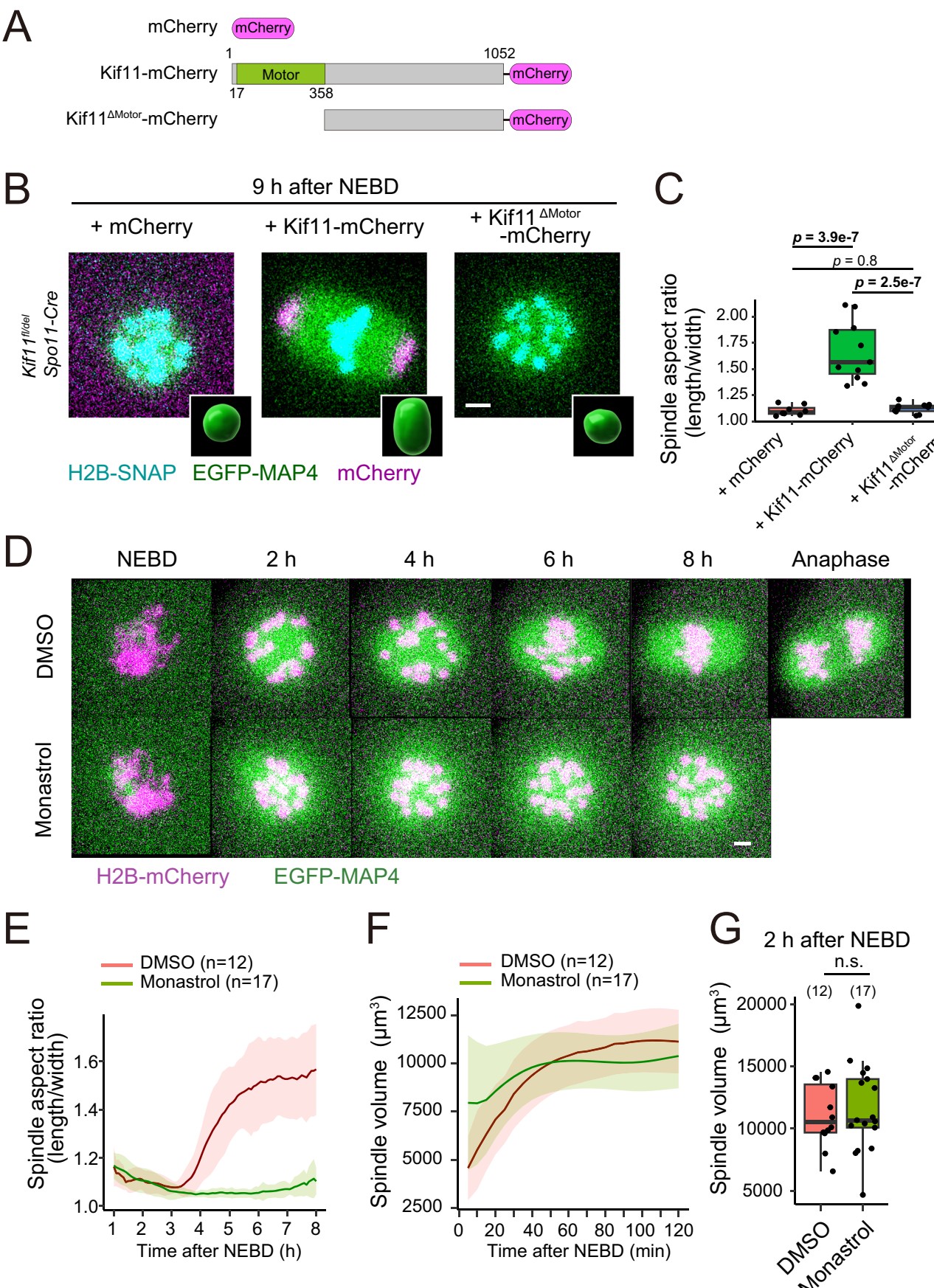

◄ **Figure EV3. Rescue experiments and phenotypes caused by KIF11 inhibition with monastrol.**

(A) Kif11 constructs used for rescue experiments. $Kif11^{\Delta Motor}$ lacks the motor domain. (B) Live imaging of $Kif11^{fl/del}$ Spo11-Cre (conditional homozygous deletion) oocytes expressing EGFP-MAP4 (microtubules, green), H2B-SNAP (chromosomes, cyan), and one of the Kif11-mCherry, $Kif11^{\Delta Motor}$-mCherry or mCherry (magenta). Z-projection images and 3D-reconstructed spindle images are shown. Scale bars, 5 μm. (C) Box plot comparing spindle elongation at metaphase (9 h after NEBD) (Welch's $t$-test comparing all possible pairs with Holm's correction for multiple comparisons). (D) Live imaging of BDF1 mouse oocytes expressing EGFP-MAP4 (microtubules, green) and H2B-mCherry (chromosomes, magenta) in the presence of DMSO (control) or monastrol. Z-projection images are shown. Scale bars, 5 μm. (E) Spindle elongation over time is shown as mean ± SD. The aspect ratio (length/width) of 3D-reconstructed spindles was measured. (F) Spindle volume with mean ± SD determined from 3D-reconstructed images. (G) Box plot comparing spindle volume at prometaphase (2 h after NEBD) between control (DMSO) and monastrol ($p = 0.63$, Welch's $t$-test). The numbers in parentheses above the box plot indicate the number of oocytes used for experiments. Three or two biologically independent experiments were performed for panels (A–G), respectively. Boxplots: Centerlines indicate the median; box limits represent the 25th and 75th percentiles; whiskers extend to the minimum and maximum values no further than 1.5 * IQR from the hinge (where IQR is the inter-quartile range, or distance between the first and third quartiles). Source data are available online for this figure.

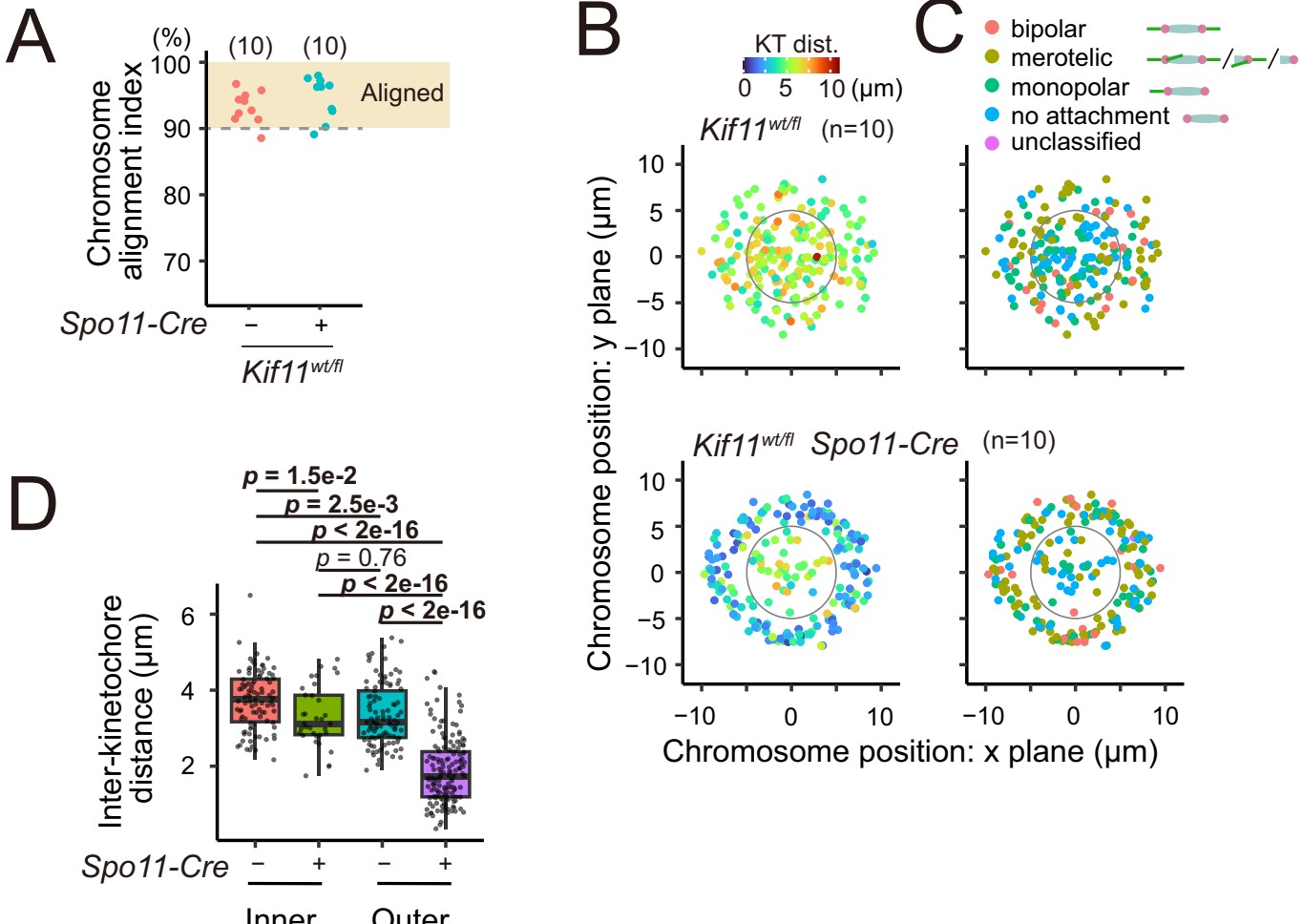

**Figure EV4. Spatial difference of KT-MT attachment status with respect to chromosome stretching.**

(A) Chromosome alignment index (see Fig. EV2) of oocytes analyzed for KT-MT attachments 5 h after NEBD. Each point represents one oocyte. The numbers in parentheses above the plots indicate the number of oocytes used for experiments. (B) Distribution of chromosome positions on the spindle equator with color codes indicating inter-kinetochore distance. (C) Chromosome positions are shown as in B, with color indicating KT-MT attachment status. (D) Box plot comparing inter-kinetochore distance with respect to chromosome position (inner or outer; see Fig. 4F) between genotypes. The statistically significant differences in inter-kinetochore distance between the chromosome positions were compared for each genotype by Welch's *t*-test for all possible pairs with Holm's correction for multiple comparisons. Three biologically independent experiments examining a total of ten oocytes for each genotype were performed. Boxplots: Centerlines indicate the median; box limits represent the 25th and 75th percentiles; whiskers extend to the minimum and maximum values no further than 1.5 * IQR from the hinge (where IQR is the inter-quartile range, or distance between the first and third quartiles). Source data are available online for this figure.

