## [Peer Review File · EMBO Reports]

Kif11-haploinsufficient oocytes reveal spatially differential requirements for chromosome biorientation in the spindle

Tappei Mishina, Aurélien Courtois, Shuhei Yoshida, Kohei Asai, Hiroshi Kiyonari, and Tomoya Kitajima

Corresponding author(s): Tomoya Kitajima (tomoya.kitajima@riken.jp)

Review Timeline:

Submission Date:	24th Dec 24
Editorial Decision:	19th Feb 25
Revision Received:	16th May 25
Editorial Decision:	1st Jul 25
Revision Received:	2nd Jul 25
Accepted:	15th Jul 25

Editor: Deniz Senyilmaz Tiebe

Transaction Report:

Dear Dr. Kitajima,

Thank you for submitting your research manuscript to our journal, which was now seen by three referees, whose reports are copied below. Please accept my apologies for the delay in getting back to you, it took longer than anticipated to receive the referee reports.

Referees, especially referees #2 and #3, express interest in the study revealing the differential Kif11 requirements of chromosomes in a manner that is dependent on their localization. However, they also raise some concerns that need to be addressed to consider publication here.

Given these positive recommendations, we would like to invite you to revise your manuscript with the understanding that the referee concerns (as in their reports) must be fully addressed and their suggestions taken on board. Please address all referee concerns in a complete point-by-point response. Acceptance of the manuscript will depend on a positive outcome of a second round of review. It is EMBO reports policy to allow a single round of major experimental revision only and acceptance or rejection of the manuscript will therefore depend on the completeness of your responses included in the next, final version of the manuscript.

We realize that it is difficult to revise to a specific deadline. In the interest of protecting the conceptual advance provided by the work, we recommend a revision within 3 months. Please discuss the revision progress ahead of this time with me if you require more time to complete the revisions, or if you have questions or comments regarding the revision (also by video chat).

1. A data availability section providing access to data deposited in public databases is missing (where applicable).
2. Your manuscript contains statistics and error bars based on $n=2$. Please use scatter plots in these cases.

You can submit the revision either as a Scientific Report or as a Research Article. For Scientific Reports, the revised manuscript can contain up to 5 main figures and 5 Expanded View figures, and it should not exceed 27000 characters. If the revision leads to a manuscript with more than 5 main figures it will be published as a Research Article. In this case the Results and Discussion section should be separate. If a Scientific Report is submitted, these sections have to be combined. This will help to shorten the manuscript text by eliminating some redundancy that is inevitable when discussing the same experiments twice. In either case, all materials and methods should be included in the main manuscript file.

4) a .docx formatted letter INCLUDING the reviewers' reports and your detailed point-by-point responses to their comments. As part of the EMBO publication's Transparent Editorial Process, EMBO reports publishes online a Review Process File (RPF) to accompany accepted manuscripts. This File will be published in conjunction with your paper and will include the referee reports, your point-by-point response and all pertinent correspondence relating to the manuscript. <https://www.embopress.org/page/journal/14693178/authorguide#transparentprocess>

5) a complete author checklist, which you can download from our author guidelines <https://www.embopress.org/page/journal/14693178/authorguide>. Please insert information in the checklist that is also reflected in the manuscript. The completed author checklist will also be part of the RPF.

6) Please note that all corresponding authors are required to supply an ORCID ID for their name upon submission of a revised manuscript (<<https://orcid.org/>>). Please find instructions on how to link your ORCID ID to your account in our manuscript tracking system in our Author guidelines <<https://www.embopress.org/page/journal/14693178/authorguide#authorshipguidelines>>

7) Before submitting your revision, primary datasets produced in this study need to be deposited in an appropriate public database (see <https://www.embopress.org/page/journal/14693178/authorguide#datadeposition>). Please remember to provide a reviewer password if the datasets are not yet public. The accession numbers and database should be listed in a formal "Data Availability" section placed after Materials & Method (see also <https://www.embopress.org/page/journal/14693178/authorguide#datadeposition>). Please note that the Data Availability Section is restricted to new primary data that are part of this study. * Note - All links should resolve to a page where the data can be accessed. *
If your study has not produced novel datasets, please mention this fact in the Data Availability Section.

Additional information on source data and instruction on how to label the files are available:
<https://www.embopress.org/page/journal/14693178/authorguide#sourcedata>

9) Our journal encourages inclusion of *data citations in the reference list* to directly cite datasets that were re-used and obtained from public databases. Data citations in the article text are distinct from normal bibliographical citations and should directly link to the database records from which the data can be accessed. In the main text, data citations are formatted as follows: "Data ref: Smith et al, 2001" or "Data ref: NCBI Sequence Read Archive PRJNA342805, 2017". In the Reference list, data citations must be labeled with "[DATASET]". A data reference must provide the database name, accession number/identifiers and a resolvable link to the landing page from which the data can be accessed at the end of the reference. Further instructions are available at <http://www.embopress.org/page/journal/14693178/authorguide#referencesformat>

- the name of the statistical test used to generate error bars and P values,
- the number (n) of independent experiments (please specify technical or biological replicates) underlying each data point,
- the nature of the bars and error bars (s.d., s.e.m.),
- If the data are obtained from n Program fragment delivered error `Can't locate object method "less" via package "than" (perhaps you forgot to load "than"?) at //ejpvfs23/sites23b/embor_www/letters/embor_decision_revise_and_review.txt line 56.' 2, use scatter blots showing the individual data points.

12) Please also note our reference format:

13) All Materials and Methods need to be described in the main text using our 'Structured Methods' format, which is required for all research articles. According to this format, the Methods section includes a Reagents and Tools Table (listing key reagents, experimental models, software and relevant equipment and including their sources and relevant identifiers) followed by a Methods and Protocols section describing the methods using a step-by-step protocol format. The aim is to facilitate adoption of the methodologies across labs. More information on how to adhere to this format as well as a downloadable template (.docx) for the Reagents and Tools Table can be found in our author guidelines:

I look forward to seeing a revised version of your manuscript when it is ready. Please let me know if you have questions or comments regarding the revision.

Kind regards,

Deniz Senyilmaz Tiebe

Deniz Senyilmaz Tiebe, PhD
Senior Scientific Editor
EMBO Reports

Referee #1:

Bipolar spindle assembly is a key process for correct chromosome segregation. Oocytes lose centrosomes and hence assemble bipolar spindle via an inside-out mechanism. The role of Kif11/Eg5 in spindle bipolarity has been extremely well established from prior work in *Xenopus* egg extracts, from RNAi in cell culture and from the use of inhibitors in mouse oocytes. Authors re-investigate this role using a genetic approach of conditional Kif11/Eg5 invalidation in mouse oocytes. They show that even half a dose of Kif11/Eg5 impacts bipolar spindle assembly. The results they present are not so surprising based on previous literature. I am not totally convinced by the novelty of the findings reported here, not fully convinced of the approach.

Comments:

1/ Why not use a ZP3-Cre promoter to produce an invalidation of Kif11/Eg5 much later, during oocyte growth rather than so early in the fetal gonad with the use of the Spo11-Cre line? Kif11/Eg5 could have other roles before oocyte resume meiosis in the primordial or secondary follicles. The best way to test whether the delay in spindle bipolarization presented in Fig2 is not due to a cumulative effect of roles before meiosis resumption as well as a role in spindle elongation would be to produce a rescue experiment in the full KO context.

2/ I have a difficulty to understand the conclusion from authors on the "concentric pattern of Kif11/Eg5 dosage for chromosome biorientation". It seems rather obvious that the density of MTs is more important inside the MTs ball than at its periphery and the differential pulling and the difference in inter-kinetochore distance, but this is an indirect consequence of the prolonged microtubule ball state and elongation delay. Maybe authors should explain a bit better why they insist on this?

3/ minor detail: authors should definitively use the Kif11/Eg5 nomenclature since Eg5 is extremely well-known for its major role in spindle bipolarization.

Referee #2:

Mishina et al. hand in a manuscript on Kif11 haploinsufficient oocytes that reveal spatially differential requirements for chromosome biorientation in the spindle.

The study is based on sophisticated genetics in mice, whose oocytes develop with different Kif11 levels over time. Wt mice are compared with mice carrying a wt and a ko allele, as well as a meiosis-specific allele that can be conditionally switched off. The latter allows to compare global loss in the entire organism to specific loss in female reproductive cells. The authors validate

expression in oocytes and show a gradual decrease in oocytes ranging from ca. 60 to 22 or only 3 % residual transcript levels, backed-up by immunofluorescence detection of protein loss.

Using this tool-box, the authors survey meiotic spindles and try to deduce the function of Kif11 from appearing defects. While spindle volume is mildly but significantly affected, the elongation of the MI spindle after initial production of MT and organization into a spherical MT ball turns out to be very sensitive to Kif11 reduction. Consistently, spindles forming with reduced Kif11 activity rarely go into anaphase. The loss of a true bipolar organization is further demonstrated using Cep192 as a marker whose pole-sharp localization is only seen upon robust Kif11 activity. Lastly, it is convincingly shown that the regular distribution of chromosomes on the spindle equator is lost when Kif11 activity is reduced, suggesting the loss of biorientation of the homologous pairs.

The manuscript is at this point already in a very mature state and I suggest publication essentially in its present form in EMBO Reports. There is one major point and a couple of minor issues that should be addressed prior to publication:

Major:

1. The authors should clearly state - throughout abstract and introduction - the special situation of meiosis I that they have investigated - as compared to MII or mitosis- in terms of bivalent attachment / tension / holding homologous pairs together. The way the manuscript reads now sounds like everything seen here can be generally assumed to be true for spindle elongation. This is by no means clear, i.e. when centriole-bases MTOCs come back, the situation may / will be different.
2. Related to 1: the authors should give an interpretation and explicitly state if they think what they see in MI applies to spindles in general (speculate about Kif11 requirements beyond MI), and, if yes or no, why that is.

Minor:

3. Fig. 3C: the graph shows a large amount / majority of the Cep192 signal stuck in a central (metaphase plate) position, which is not reflected in the images shown.
4. Figure legend 1: (D) instead of the second (C).

Referee #3:

The manuscript by Mishina and colleagues describes the consequences of reduced Kif11 levels on spindle formation and chromosome biorientation in the first meiotic division in mouse oocytes. For this, they have several mouse strains with distinct Kif11 levels. Reduction of Kif11 leads to proportional failures in spindle bipolarization. Importantly, chromosome biorientation is affected, but foremost for chromosomes that are at the outer region of the metaphase plate. Overall, the study is well executed, but there are two points that should be addressed:

- 1) The authors conclude that lowering Kif11 levels leads to the observed phenotypes. If this is true, the phenotypes should be rescued by expressing wild type Kif11, but not mutant Kif11. This control is missing.
- 2) the result that chromosomes are less stretched in the outer region of the metaphase plate, than inside is very interesting. However, the authors should show attachment status of kinetochores (are there more merotelic attachment in the outer region such as expected if biorientation is affected? Or is it the spindle tension that is lower in the outer region?). Super resolution images of fixed oocytes, showing KT distances on the differently stretched chromosomes, or their attachment status would be helpful.

Point-by-point response to reviewer's comments

EMBOR-2024-61077V1

Kif11-haploinsufficient oocytes reveal spatially differential requirements for chromosome biorientation in the spindle

Correspondence to: tomoya.kitajima@riken.jp

Plain letters indicate reviewers' comments.

Blue letters indicate our responses.

"Red letters enclosed in quotation marks" indicate texts newly added to the revised manuscript.

First of all, we thank all reviewers for their suggestions and comments, which greatly improved our manuscript.

Referee #1:

Bipolar spindle assembly is a key process for correct chromosome segregation. Oocytes lose centrosomes and hence assemble bipolar spindle via an inside-out mechanism. The role of Kif11/Eg5 in spindle bipolarity has been extremely well established from prior work in *Xenopus* egg extracts, from RNAi in cell culture and from the use of inhibitors in mouse oocytes. Authors re-investigate this role using a genetic approach of conditional Kif11/Eg5 invalidation in mouse oocytes. They show that even half a dose of Kif11/Eg5 impacts bipolar spindle assembly. The results they present are not so surprising based on previous literature. I am not totally convinced by the novelty of the findings reported here, not fully convinced of the approach.

Comments:

1/ Why not use a ZP3-Cre promoter to produce an invalidation of Kif11/Eg5 much later, during oocyte growth rather than so early in the fetal gonad with the use of the Spo11-Cre line? Kif11/Eg5 could have other roles before oocyte resume meiosis in the primordial or secondary follicles. The best way to test whether the delay in spindle bipolarization presented in Fig2 is not due to a cumulative effect of roles before meiosis resumption as well as a role in spindle elongation would be to produce a rescue experiment in the full KO context.

Thank you for your question. We used *Spo11-Cre* because it depleted Kif11 RNA more efficiently compared to *Zp3-Cre* or *Gdf9-Cre*. The revised manuscript now shows this data (**revised Fig. 1D and new Fig. EV1B**) with the following statements.

"Comparable levels of depletion were achieved with *Gdf9-Cre* or *Zp3-Cre* (Fig. 1D), two other widely used oocyte-specific drivers (Lewandoski et al, 1997; Vries et al., 2000; Lan et

al, 2004). However, further analysis showed that the use of *Gdf9*-Cre or *Zp3*-Cre resulted in residual detectable reads corresponding to the last exon of the *Kif11* gene, whereas *Spo11*-Cre greatly reduced them (Fig. EV1B). We therefore decided to use *Spo11*-Cre for conditional gene knockout in this study." (L109–114)

We agree with the reviewer that a rescue experiment is critical. The revised manuscript now shows that microinjection of *Kif11* mRNA to *Kif11^{fl/del} Spo11-Cre* oocytes (the full KO context) rescued spindle elongation during meiosis (**new Figs. EV3A–C**).

"Importantly, the spindle elongation defect was rescued by expressing full-length KIF11, but not the mutant form of KIF11 lacking its motor domain, by RNA microinjection into fully grown oocytes (Fig. EV3A–C)" (L146–148)

I have a difficulty to understand the conclusion from authors on the "concentric pattern of *Kif11*/*Eg5* dosage for chromosome biorientation". It seems rather obvious that the density of MTs is more important inside the MTs ball than at its periphery and the differential pulling and the difference in inter-kinetochore distance, but this is an indirect consequence of the prolonged microtubule ball state and elongation delay. Maybe authors should explain a bit better why they insist on this?

Thank you for your question. We apologize that our statement was unclear. First, we would like to clarify that we are not claiming that *Kif11*/*Eg5* dosage itself shows a concentric pattern, but rather that the requirement for *Kif11*/*Eg5* dosage shows a concentric pattern. The original manuscript mentioned "concentric pattern of KIF11 dosage dependency for chromosome biorientation on the spindle equator", but this statement could have been misleading. We have revised this statement to:

"KIF11 insufficiency reveals spatially differential requirement for homologous chromosome biorientation on the spindle equator." (L176–177)

Second, we apologize if we misunderstood your comment, but we understand that you are suggesting that the higher density of microtubules inside the microtubule ball is important for spatial difference in chromosome biorientation. We agree with this view and have added the following sentence for clear statement.

"It is possible that the acentrosomal spindle has denser microtubules in its inner region prior to bipolarization, which may subsequently promote KIF11-mediated chromosome biorientation." (L227–229)

This hypothesis is consistent with what we discussed in our original manuscript: concentrated chromosome-derived signals inside the spindle may create spatial differences in chromosome

biorientation (L230–237).

Furthermore, our idea is consistent with your idea that the spatial differences in chromosome biorientation are a result of the prolonged microtubule ball state and elongation delay. We suggest that chromosome biorientation in oocytes is inherently spatially different, and that *Kif11* haploinsufficiency resulted in a phenotype in which the spatial difference was clearly visible. To clarify this idea, we have revised our statement:

“The delay in spindle elongation in *Kif11*-haploinsufficient oocytes resulted in chromosome biorientation defects, particularly at the outer region of the spindle equator. These results suggest a spatially differential requirement for homologous chromosome biorientation on the spindle equator in oocytes” (L212–214)

Finally, we understand that the reviewer’s concern may stem from a lack of clarity about what is actually different between the inner and outer regions of the spindle equator. The revised manuscript now includes new data showing that the spindle prefers to form merotelic kinetochore-microtubule attachment in the outer region (**new Figs. 4E,F**). We suggest that this spatial difference makes chromosome stretching in the outer region particularly challenging and therefore sensitive to a reduction in KIF11 dosage (**L247–251**). These new data and discussions have significantly improved our manuscript.

We appreciate your helpful comments and hope that our revisions have addressed your concerns.

3/ minor detail: authors should definitively use the *Kif11*/*Eg5* nomenclature since *Eg5* is extremely well-known for its major role in spindle bipolarization.

Thank you for pointing out this. The revised manuscript now clearly states this.

“The kinesin motor KIF11 (also widely known as *Eg5*)” (L19)

Referee #2:

Mishina et al. hand in a manuscript on Kif11 haploinsufficient oocytes that reveal spatially differential requirements for chromosome biorientation in the spindle.

The study is based on sophisticated genetics in mice, whose oocytes develop with different Kif11 levels over time. Wt mice are compared with mice carrying a wt and a ko allele, as well as a meiosis-specific allele that can be conditionally switched off. The latter allows to compare global loss in the entire organism to specific loss in female reproductive cells. The authors validate expression in oocytes and show a gradual decrease in oocytes ranging from ca. 60 to 22 or only 3 % residual transcript levels, backed-up by immunofluorescence detection of protein loss.

Using this tool-box, the authors survey meiotic spindles and try to deduce the function of Kif11 from appearing defects. While spindle volume is mildly but significantly affected, the elongation of the MI spindle after initial production of MT and organization into a spherical MT ball turns out to be very sensitive to Kif11 reduction. Consistently, spindles forming with reduced Kif11 activity rarely go into anaphase. The loss of a true bipolar organization is further demonstrated using Cep192 as a marker whose pole-sharp localization is only seen upon robust Kif11 activity. Lastly, it is convincingly shown that the regular distribution of chromosomes on the spindle equator is lost when Kif11 activity is reduced, suggesting the loss of biorientation of the homologous pairs.

The manuscript is at this point already in a very mature state and I suggest publication essentially in its present form in EMBO Reports. There is one major point and a couple of minor issues that should be addressed prior to publication:

Major:

1. The authors should clearly state - throughout abstract and introduction - the special situation of meiosis I that they have investigated - as compared to MII or mitosis- in terms of bivalent attachment / tension / holding homologous pairs together. The way the manuscript reads now sounds like everything seen here can be generally assumed to be true for spindle elongation. This is by no means clear, i.e. when centriole-bases MTOCs come back, the situation may / will be different.

Thank you for this suggestion. We fully agree with the reviewer. The revised manuscript now carefully states the special situations of meiosis I in oocytes throughout abstract and introduction. The revised statements include the following:

“...elongating a spherical spindle into a bipolar-shaped structure in **acentrosomal** oocytes. **During meiosis I**, this process stretches **homologous chromosome pairs**, establishing

chromosome biorientation at the spindle equator. The quantitative requirement for KIF11 in **acentrosomal** spindle bipolarization and **homologous** chromosome biorientation remains unclear.” (Abstract, L20–24)

“These findings demonstrate the necessity of biallelic functional Kif11 for bipolar spindle assembly in **acentrosomal** oocytes and reveal a spatially differential requirement for **homologous** chromosome biorientation within the spindle.” (Abstract, L29–32)

“In mammalian oocytes, which lack canonical centrosomes, entry to the M-phase of meiosis I promotes...” (Introduction, L43)

“Spindle elongation allows microtubules to stretch **homologous** chromosome **pairs** toward the opposite poles” (Introduction, L54–56)

“whether a heterozygous loss-of-function mutation of *Kif11* impairs **acentrosomal** spindle bipolarization or **homologous** chromosome biorientation in oocytes remains unknown.” (Introduction, L79–81)

2. Related to 1: the authors should give an interpretation and explicitly state if they think what they see in MI applies to spindles in general (speculate about Kif11 requirements beyond MI), and, if yes or no, why that is.

Thank you for this suggestion. We favor the idea that our findings in this study apply to oocyte MI rather than to spindles in general. The revised manuscript clearly states this, and why we think so:

“Being occupied and surrounded by chromosomes, the inner region may experience higher concentrations of chromosome-derived signals, creating a microenvironment that enhances KIF11-mediated chromosome biorientation. For instance, chromosomes produce diffusible RanGTP signals that activate spindle assembly factors (Carazo-Salas et al, 1999; Kalab et al, 2002) such as HURP, a bundling factor that localizes to kinetochore-attached microtubules (Silljé et al, 2006; Wong & Fang, 2006; Koffa et al, 2006). HURP accumulates more prominently in the inner region of the spindle equator (Takenouchi et al, 2024), potentially cooperating with KIF11 to facilitate chromosome biorientation. **Given that chromosomal RanGTP signals are more essential for spindle assembly in acentrosomal oocytes than in centrosomal somatic cells (Dumont et al, 2007; Schuh & Ellenberg, 2007; Moutinho-Pereira et al, 2013; Holubcová et al, 2015; Drutovic et al, 2020), the spatially differential nature of chromosome biorientation may be specific to acentrosomal oocytes. It is also possible that the meiosis I-specific chromosome feature contributes to the spatially differential nature of biorientation, as homologous chromosome biorientation during meiosis I requires kinetochore pair separation over greater distances compared to sister chromatid biorientation during meiosis II and mitosis.” (L230–245)**

Minor:

3. Fig. 3C: the graph shows a large amount / majority of the Cep192 signal stuck in a central (metaphase plate) position, which is not reflected in the images shown.

Thank you for this comment. As you pointed out, a large amount of Cep192 signals stuck in a central position in *Kif11^{fl/del}* oocytes in Fig. 3C, which was not reflected in the images shown in Fig. 3A. We went to the original datasets and found that *Kif11^{fl/del}* oocytes could be categorized into two types – one showing a relatively good Cep192 bipolar sorting (which was reflected in the original Fig. 3A) and the other showing a severe Cep192 stuck in a central position (which was apparent in Fig. 3C but not reflected in the original Fig. 3A). We decided to show both types of oocytes in the revised figure panel (**revised Fig. 3A**).

4. Figure legend 1: (D) instead of the second (C).

Thank you for finding this out. Corrected.

Referee #3:

The manuscript by Mishina and colleagues describes the consequences of reduced Kif11 levels on spindle formation and chromosome biorientation in the first meiotic division in mouse oocytes. For this, they have several mouse strains with distinct Kif11 levels. Reduction of Kif11 leads to proportional failures in spindle bipolarization. Importantly, chromosome biorientation is affected, but foremost for chromosomes that are at the outer region of the metaphase plate. Overall, the study is well executed, but there are two points that should be addressed:

1) The authors conclude that lowering Kif11 levels leads to the observed phenotypes. If this is true, the phenotypes should be rescued by expressing wild type Kif11, but not mutant Kif11. This control is missing.

Thank you for suggesting the very important experiment. The revised manuscript now shows that the introduction of Kif11 mRNA to *Kif11^{fl/del} Spo11-Cre* oocytes (the full KO context) at the GV stage rescued spindle elongation during meiosis. This rescue was not observed with the mutant form of Kif11 that lacks the motor domain (**new Fig. EV3A–C**).

“Importantly, the spindle elongation defect was rescued by expressing full-length KIF11, but not the mutant form of KIF11 lacking its motor domain, by RNA microinjection into fully grown oocytes (Fig. EV3A–C)” (L146–148)

2) the result that chromosomes are less stretched in the outer region of the metaphase plate, than inside is very interesting. However, the authors should show attachment status of kinetochores (are there more merotelic attachment in the outer region such as expected if biorientation is affected? Or is it the spindle tension that is lower in the outer region?). Super resolution images of fixed oocytes, showing KT distances on the differently stretched chromosomes, or their attachment status would be helpful.

Thank you for this important question. To address this, we visualized kinetochore-microtubule attachments in control and heterozygous *Kif11* KO oocytes. Interestingly, in control oocytes, where all chromosomes were fully stretched in both the outer and inner regions, tended to show more merotelic kinetochore-microtubule attachments in the outer region compared to the inner region (**new Figs. 4E,F**). Importantly, this spatial tendency of attachment status was also observed in heterozygous *Kif11* KO oocytes, where most chromosomes are unstretched in the outer region (**new Figs. 4E,F**). These observations suggest that the spindle prefers to form merotelic kinetochore-microtubule attachments in the outer region of its equator, regardless of the stretched or unstretched state of chromosomes. We speculate that the full dose of KIF11 is required for chromosomes with merotelic attachments to stretch in the outer region. The revised manuscript includes the following statements:

“To gain insight into the spatial difference in the requirement for chromosome stretching, we visualized kinetochore-microtubule attachments (Fig. 4E). In control oocytes, where all chromosomes were aligned and fully stretched in both the outer and inner regions of the spindle equator (Figs. EV4A–D), kinetochores attached to microtubules from opposite poles (merotelic attachment) were preferentially found in the outer region (Fig. 4F). Notably, a similar spatial bias of merotelic attachment to the outer region was observed in heterozygous conditional deletion oocytes (*Spo11-Cre, Kif11^{wt/fl}*) (Fig. 4F), where most chromosomes were not stretched in the outer region (Figs. EV4A–D). These results suggest that the spindle prefers to form merotelic kinetochore-microtubule attachments in the outer region of the equator, regardless of the stretched or unstretched state of chromosomes. We propose that the full dose of KIF11 ensures chromosome stretching with merotelic kinetochores in the outer region of the spindle equator.” (L196–207)

“...pulling forces are relatively weaker in the outer region, which is associated with increased opportunities for merotelic kinetochore-microtubule attachment. We suggest that this spatial difference makes chromosome stretching in the outer region particularly challenging and therefore sensitive to a reduction in KIF11 dosage.” (L247–251)

These new data and discussions have significantly improved our manuscript. We are grateful to the reviewer for his/her thoughtful suggestion.

Dear Dr. Kitajima,

Thank you for submitting your revised manuscript. It has now been seen by one of the original referees.

As you will see, referee finds that the study is significantly improved during revision and recommend publication. However, the editorial points below need to be addressed before I can accept the manuscript.

- We find that your manuscript is better suited for our 'Report' format.
- Please rename the "Competing interests" section as "Disclosure and Competing Interests Statement".
- We note the following regarding the figure callouts: Fig. 3B and 3C are currently not called out in the text. Fig. S1A is an incorrect callout and needs to be updated as Fig. EV1A.
- We note the following regarding source data: numerical data are uploaded as one Excel file with multiple sheets. However, they need to be uploaded as one zip folder per figure where each folder should have separate files/folders, one item per panel. We note that source data for 1B/D, 2AF, 3A are currently missing. Fig 4A; micr. images are stated to be in S-BIAD1946, which needs to be publicly available and a direct URL needs to be provided in the Data Availability section and in the source data checklist stating the related figure panels.
- Please make the dataset GSE284383 publicly available and remove the reviewer token from the manuscript text.
- Our production/data editors have asked you to clarify several points in the figure legends - Figure Legends (main + EV):
 - o Please note that the exact p values are not provided in the legend of figure EV4 D
 - o Please indicate the statistical test used for data analysis in the legends of figures 1D, 2F, EV1 B
 - o Please note that the box plots need to be defined in terms of minima, maxima, centre, bounds of box and whiskers, and percentile in the legends of figures 1D, 2F, EV1 B, EV3 C, G
 - o Please note that information related to n is missing in the legend of figure EV2 B
 - o Please note that scale bar and its definition are missing for figure 2A
- Papers published in EMBO Reports include a 'synopsis' and 'bullet points' to further enhance discoverability. Both are displayed on the html version of the paper and are freely accessible to all readers. The synopsis includes a short standfirst summarizing the study in 1 or 2 sentences (max 35 words) that summarize the paper and are provided by the authors and streamlined by the handling editor. I would therefore ask you to include your synopsis blurb and 3-5 bullet points listing the key experimental findings.
- In addition, please provide an image for the synopsis. This image should provide a rapid overview of the question addressed in the study but still needs to be kept fairly modest since the image size cannot exceed 550 (width) x 300-600 (height) pixels.

Thank you again for giving us to consider your manuscript for EMBO Reports, I look forward to your minor revision.

Kind regards,

Deniz Senyilmaz Tiebe

--

Deniz Senyilmaz Tiebe, PhD
Senior Scientific Editor
EMBO Reports

Referee #3:

The authors have addressed my concerns and added more experiments that greatly increase the quality of the manuscript. I think the manuscript should be published in its current form.

All editorial and formatting issues were resolved by the authors.

Dr. Tomoya Kitajima
RIKEN Center for Biosystems Dynamics Research
Laboratory for Chromosome Segregation
2-2-3 Minatojima-minamimachi, Chuo-ku
Kobe
650-0047
Japan

Dear Dr. Kitajima,

Thank you for submitting your revised manuscript. I have now looked at everything and all is fine. Therefore, I am very pleased to accept your manuscript for publication in EMBO Reports.

Congratulations on a nice work!

Kind regards,

Deniz Senyilmaz Tiebe

--

Deniz Senyilmaz Tiebe, PhD
Senior Scientific Editor
EMBO Reports
